# Genome-wide mutational signatures in low-coverage whole genome sequencing of cell-free DNA

Jonathan C. M. Wan [1], Dennis Stephens[1], Lingqi Luo[1], James R. White [1,2], Caitlin M. Stewart [1,4,5], Benoît Rousseau [1], Dana W. Y. Tsui[3,6] & Luis A. Diaz Jr. [1] ✉

Mutational signatures accumulate in somatic cells as an admixture of endogenous and exogenous processes that occur during an individual's lifetime. Since dividing cells release cell-free DNA (cfDNA) fragments into the circulation, we hypothesize that plasma cfDNA might reflect mutational signatures. Point mutations in plasma whole genome sequencing (WGS) are challenging to identify through conventional mutation calling due to low sequencing coverage and low mutant allele fractions. In this proof of concept study of plasma WGS at 0.3–1.5x coverage from 215 patients and 227 healthy individuals, we show that both pathological and physiological mutational signatures may be identified in plasma. By applying machine learning to mutation profiles, patients with stage I-IV cancer can be distinguished from healthy individuals with an Area Under the Curve of 0.96. Interrogating mutational processes in plasma may enable earlier cancer detection, and might enable the assessment of cancer risk and etiology.

Earlier detection of cancer improves the likelihood of eligibility for effective treatments such as surgery, resulting in a greater chance of survival, reduced morbidity, and less expensive treatment[1]. Liquid biopsies are increasingly being utilized for non-invasive cancer detection, prognostication, and treatment monitoring[2]. Current methods for early detection using circulating tumor DNA (ctDNA) detect features of the tumor in plasma, which can be linked to the etiology and type of cancer, such as point mutations[3,4], copy number alterations[5,6] or methylation patterns[7]. Other features in plasma may be related to the biology of cfDNA, such as fragmentation patterns of cfDNA from cancer cells[8,9].

Mutational processes are ongoing in somatic cells throughout the lifetime of an individual[10,11]. Endogenous processes (such as aging), and exogenous exposures (such as smoking) both cause distinct mutational signatures in the genomes of somatic cells[12,13]. Since dividing cells release DNA fragments into the circulation as cell-free DNA (cfDNA)[2,14], with multiple tissues represented[15], cfDNA may reflect mutational signatures of somatic tissues. By applying a personalized sequencing method, it was shown that despite the limited depth, low-coverage WGS contains point mutation signals at patient-specific loci[16].

Conventionally, somatic mutation signature extraction from cancer tissue whole-genome sequencing (WGS) has been performed on mutation calls from matched tumor and normal samples at moderate sequencing depth[12,17]. In low-coverage WGS data, somatic and germline mutations are likely to be indistinguishable by allele fraction alone, precluding the use of per-locus allele fraction-based germline filters[16,18]. Conventional mutation calling approaches, which require multiple mutant reads to support the call[19], may not be used. Even with

[1]Division of Solid Tumor Oncology, Memorial Sloan Kettering Cancer Center, New York, NY 10065, USA. [2]Resphera Biosciences, Baltimore, MD 21231, USA. [3]Department of Pathology, Memorial Sloan Kettering Cancer Center, New York, NY 10065, USA. [4]Present address: Meyer Cancer Center, Weill Cornell Medical College, New York, NY 10065, USA. [5]Present address: New York Genome Center, New York, NY 10013, USA. [6]Present address: PetDx Inc., San Diego, CA, USA. ✉e-mail: diazl5@mskcc.org

## Pointy overview

**Fig. 1 | Study outline and characterization of Pointy data.** cfDNA libraries were generated from plasma samples from patients with cancer and healthy individuals from two independent cohorts. Whole-genome sequencing (WGS) was performed to 0.3–1.5× coverage (Supplementary Data 1). Mutational signatures were extracted from these data, enabling signature profiling and sample classification ("Methods"), tested in two independent cohorts.

high sequencing depths of WGS, dilution of mutant DNA in wild-type cfDNA would still result in many true mutation loci being observed with one mutant read at best when ctDNA fractions are low[16,20]. This is due to the long tail of low allele fraction mutations in the tumor being occasionally sampled in plasma.

Here, we show an approach called Pointy, which enables the analysis of genome-wide mutational signatures from low-coverage plasma WGS (0.3–1.5× depth). We demonstrate single-base substitution (SBS) signature profiling and sample classification using a combination of signature extraction and machine learning (Fig. 1). Germline sequencing is not performed to maximize scalability, and we implement methods to mitigate technical and biological noise. We identify mutational signatures in the plasma of individuals with and without cancer, which may be leveraged for early cancer detection.

## Results

### Characterizing and normalizing Pointy data

We developed a pipeline to extract point mutations from low-coverage plasma WGS called Pointy ("Methods" and Supplementary Fig. 1). We first explored a cohort of patients with stage IV colorectal cancer (CRC, $n = 16$), many of whom had mismatch repair deficiency (MMR-D) and/or microsatellite instability[21] (MSI). Healthy controls from the same cohort were used ($n = 19$). Each library was sequenced to a median of $31.0 \times 10^6$ reads, with a median duplication rate of 0.37%. Data were downsampled to a target of 0.3× (10M paired-end reads), which resulted in a median of $10.0 \times 10^6$ reads. Samples with fewer than 90% of the target number of reads were not evaluated ($n = 2$). A median of 79.3% of genomic positions had zero coverage, and 14% of bases had 1× coverage, equating to a mean coverage of 0.28× (95% confidence interval (CI) 0.26–0.29×, Supplementary Fig. 2a).

In this study, error-suppression by read collapsing of duplicates is limited by the low duplication rate of WGS (<0.5% duplication rate, Supplementary Fig. 2b). Instead, we utilized error-suppression filters based on previous work[16,20], as follows: minimum base quality (BQ) threshold of 30, mean BQ threshold of 30, requiring mutations to be present in both read 1 (R1) and read 2 (R2), and mapping quality (MQ) threshold of 60. After applying these filters, a mean of 9886 mutations per sample was retained, prior to SNP filtering (95% CI 8782–10,990, Supplementary Fig. 2c).

The samples from the PGDX cohort were sequenced in two batches from the same sequencing instrument, so we explored data from healthy individuals for batch effects. In healthy samples, there was no significant difference in the mean number of mutations between batches (9049 vs. 10,089, $p = 0.47$, two-sided Wilcoxon test, Supplementary Fig. 3a). However, Principal Component Analysis (PCA) of SBS profiles revealed a difference in mutation profile (Supplementary Fig. 3b), which may arise from differences in GC-bias between sequencing runs (Supplementary Fig. 3c). We identified a significant difference in the mean contribution of PC2 per sample (unadjusted $p = 0.022$, two-sided Wilcoxon test, Supplementary Fig. 3d). The largest contributors to PC2 were contexts at the extremes of GC content (Supplementary Fig. 3e). Therefore, the GC bias for each sample was determined, as was the average GC profile of the sequencing batch, which was combined to normalize the SBS profile of each sample ("Methods"). This approach is analogous to GC-correction methods used to correct whole-genome copy-number[5,22] or fragmentation profiles[9]. After GC-correction, there were no significant differences in any PC between the two sequencing runs (unadjusted $p > 0.05$, two-sided Wilcoxon test, Supplementary Fig. 3f). In Supplementary Fig. 3g, we show high cosine similarities between samples even without GC-correction, which increased significantly following GC-correction (0.995 vs 0.999, $p < 2.2 \times 10^{-16}$, two-sided Wilcoxon test). Differences in SBS profile with and without GC-correction are shown in Supplementary Fig. 3h.

Following GC-bias normalization, cancer patient plasma samples and healthy controls showed SBS mutation profiles that had a cosine similarity of 0.999 (95% CI 0.999–0.999, Supplementary Fig. 4a), although this included SNPs. Samples from cancer patients showed significantly more point-mutated reads compared to healthy controls (median 11,786, vs. 9322, $p = 0.028$, two-sided Wilcoxon test, Supplementary Fig. 4b).

### Detection of mutational signatures in CRC plasma

Mutational signatures were fitted to mutation profiles after background subtraction ("Methods"), i.e. for each sample, for each SBS context, the median number of mutations in controls was subtracted ("Methods"). Sequencing artifact signatures were included in the database used for signature fitting to minimize the misattribution of mutations to biologically relevant signatures.

We assessed the sensitivity and specificity of signature fitting in silico. Between 10 and 1000 mutations belonging to each SBS signature were spiked into a randomly selected control sample, repeated 100 times (Supplementary Methods). The contribution of each signature was assessed pre- and post-spike. Across all SBS signatures, the mean sensitivity of spiking was 38% for 10 mutations and 93% for 1000 mutations (Supplementary Fig. 5a). An example control SBS profile is shown following each spike-in (Supplementary Fig. 5b). Signatures whose profiles were concentrated in few SBS contexts, such as SBS1 and SBS2, were more efficiently fitted to than flat signatures such as SBS5 (Supplementary Fig. 5c).

To assess the performance of signature recovery in the setting of multiple signatures, we iteratively spiked in signatures and simultaneously spiked in SBS1 at a ratio of 1:1 or 10:1 (Supplementary Fig. 5d and Supplementary Methods). At a 1:1 ratio of spike-in of both signatures, there was no impact on signature fitting. However, when 10×

more SBS1 mutations were spiked in compared to the signature of interest, the rate of on-target signature fitting was reduced in multiple signatures (Benjamini−Hochberg corrected $p < 0.05$), especially in signatures with low cosine similarity to SBS1 (linear regression $p = 1.5 \times 10^{-9}$). In contrast, signatures with high similarity to SBS1 gained mutations directly from SBS1 ($q > 0.05$). We show the extent of false positive signature fitting in the context of a singly spiked signature in Supplementary Fig. 6, where the proportion of mutations that were misattributed ranged from 1.7% with 10 mutations spiked, to 0.1% with 1000 mutations spiked.

In CRC samples, the largest contributor to plasma Pointy signatures were SBS1 (aging) and SBS54 (probable SNP contamination), which comprised a median of 339 (13.0%) and 379 (15.0%) mutations, respectively (Fig. 2a and Supplementary Data 2). Compared to healthy individuals, CRC patient plasma showed significantly greater contributions of multiple signatures, including SBS1 and SBS21 (Benjamini−Hochberg (BH) adjusted $p = 0.036$, one-sided Wilcoxon test, Fig. 2a). The latter is consistent with previously detected microsatellite instability (MSI) in these patients[21].

Given the role of aging and MSI in this CRC cohort, we studied aging and MSI signatures, including SBS1, SBS5, SBS20, and SBS21. Both aging and MSI signatures had significantly higher contributions in the plasma of patients with CRC (Fig. 2a, b), and remained significant when iteratively downsampled to 10M reads 50 times (Supplementary Fig. 7). We tested whether these plasma signature contributions correlated with both ctDNA fraction and tumor mutation burden (TMB). ctDNA fraction was determined by ichorCNA[5] and tumor mutation burden was determined by targeted panel sequencing of plasma[21]. Multiple aging and MSI-associated signatures showed a significant correlation with ctDNA fraction (Fig. 2c) and TMB (adjusted $p \le 0.05$, Fig. 2d).

To detect individual signatures per sample, signature detection was performed ("Methods"). The healthy samples were used as a panel of normals, with a threshold of 95% specificity for detection of each signature. Aging signatures were detected in 10 out of 16 (62.5%) patients, and MSI signatures in 11 out of 16 (69.0%, Fig. 2e). Patients with MSI-H tumors had significantly greater SBS20 and SBS21 contributions than controls, whereas patients with MSS tumors were non-significantly different (Supplementary Fig. 8).

Signatures identified in CRC patient samples were compared against signatures fitted to targeted sequencing mutation calls on the same samples[21] (Supplementary Methods). Both approaches identified aging and MSI signatures, with 77.6% agreement across all signatures (Supplementary Fig. 9). Targeted sequencing identified SBS15 (Supplementary Fig. 9a), which was not detected with 95% specificity in Pointy data. We suggest that SBS15 mutations may have been misattributed to SBS1 given their high cosine similarity (Supplementary Fig. 9b), combined with the relatively low sensitivity of Pointy for SBS15 from spike-in benchmarking (Supplementary Fig. 5a). When the cluster of similar signatures identified in Supplementary Fig. 9b (namely, SBS1 and SBS6) were excluded from signature fitting, SBS15 could be observed (Supplementary Fig. 9c−e). Germline subtraction and mutation calling would likely improve the resolution of signature profiling, although this would conventionally require 1−2 orders of magnitude greater sequencing[17,20].

**SNP subtraction and signature fitting.** Signature fitting was repeated on the same Pointy data with SNP subtraction. Following SNP subtraction, SBS1´ (SBS1´ = SBS1 with SNP subtraction) and SBS5´ were assigned a median of zero mutations each (Supplementary Fig. 10a and Supplementary Data 3). We hypothesized that the SNP database contained aging mutations, which had been subtracted from Pointy data. To assess the bias introduced by SNP-subtraction, the SBS profile of the aggregated SNPs from the 1000 Genomes database[23] was generated (Supplementary Fig. 10b). We confirmed that the majority

of SNPs fitted SBS1 (12.1%) and SBS5 (63.2%, Supplementary Fig. 10c). Despite the mutation profile bias introduced by SNP subtraction, removal of mutated reads at SNP sites reduced the cosine similarity between the SBS profiles of cases and controls to a mean of 0.982 using bootstrapping with 50 iterations (95% CI 0.982−0.983, Supplementary Fig. 10d), compared to 0.999 when SNPs were retained (95% CI 0.999−0.999, Supplementary Fig. 4a). Together, these data suggest that SNP-subtracted data may be more suited to cancer classification, whereas SNP-retained data may provide a less biased profile of fitted signatures.

### Colorectal cancer detection

We next sought to classify samples into cancer vs. healthy based on their SBS mutation profile. To maximize the signal-to-noise ratio, SNPs were subtracted. Then, SBS´ (SNP-subtracted) mutation profiles underwent dimensionality reduction using Principal Component Analysis (PCA), and the principal components of SBS profiles (analogous to mutational signatures) were used for machine learning classification (Methods).

PCA showed separation of cases and controls based on two Principal Components, particularly in PC2 (Fig. 3a). As SNP-subtracted data were used, few mutations were fitted to aging signatures due to the bias introduced by SNP-subtraction (Supplementary Fig. 10a). PC1 and PC2 from SNP-subtracted data were both significantly correlated with ctDNA fraction ($p < 0.0076$, Fig. 3b). The signature contributions to PC1 and PC2 were assessed, which showed SBS8´ was the greatest contributor to PC2 (Fig. 3c). The SBS1 signature contribution was significantly correlated with SBS8´ (Pearson $r = 0.63$, $p = 5.6 \times 10^{-5}$, Fig. 3d), suggesting aging mutations were being fitted to SBS8´ when SNPs were subtracted.

To classify samples as either cancer or healthy, we used SBS mutation profiles as input to a machine learning model. Four methods were tested, including xgboost[24], random forest (RF), support vector machine (SVM), and logistic regression. For each condition, nested ten-fold cross-validation was performed, repeated 10 times ("Methods"). First, we assessed whether raw SNP-subtracted 96-SBS profiles could be used as input to the xgboost model, or if dimensionality should be reduced. Raw mutation input resulted in an AUC of 0.82 (95% CI 0.71−0.90, Supplementary Fig. 11a), whereas PCA-transformed input gave an AUC of 0.92 (95% CI 0.88−0.97, Supplementary Fig. 11b). Across all models, with SNPs subtracted, a median AUC of 0.96 was reached (range 0.92−0.98, Supplementary Fig. 11b−e), with the RF model performing best (AUC 0.98, 95% CI 0.90−1.00). Adding ichor ctDNA fraction to the model improved the AUC of the RF model to 1.00 (95% CI 1.00−1.00, Supplementary Fig. 11f), which was selected for subsequent analyses. We confirmed this result with ten-fold cross-validation, repeated 500 times (AUC 0.99, 95% CI 0.95−1.00, Fig. 3e). To assess the effect of downsampling, we iteratively downsampled the data to 10M reads 50 times, confirming a mean AUC of 0.97 (95% CI 0.90−1.00, Supplementary Fig. 11g).

To confirm the enhanced signal-to-noise ratio following removal of SNPs from Pointy data, classification was performed using RF with SNPs retained, which showed an AUC of 0.65 (95% CI 0.53−0.76, Supplementary Fig. 11h). We also quantified the effect of error-suppression, i.e., requiring mutations to be supported in both F and R mate pairs vs. being supported in either F or R only. This showed a significant increase in AUC associated with error suppression (0.93 without vs. 0.98 with error-suppression, $p = 0.004$, Wilcoxon test, Supplementary Fig. 11d, i). Therefore, for subsequent analyses for cancer detection, we processed data by using (a) SNP-subtraction, (b) PC-transformation, (c) error-suppression, and (d) detection using an RF model ("Methods").

**Signature detection in plasma across multiple cancer types.** We next applied Pointy to the Cristiano et al.[9] plasma WGS dataset to test

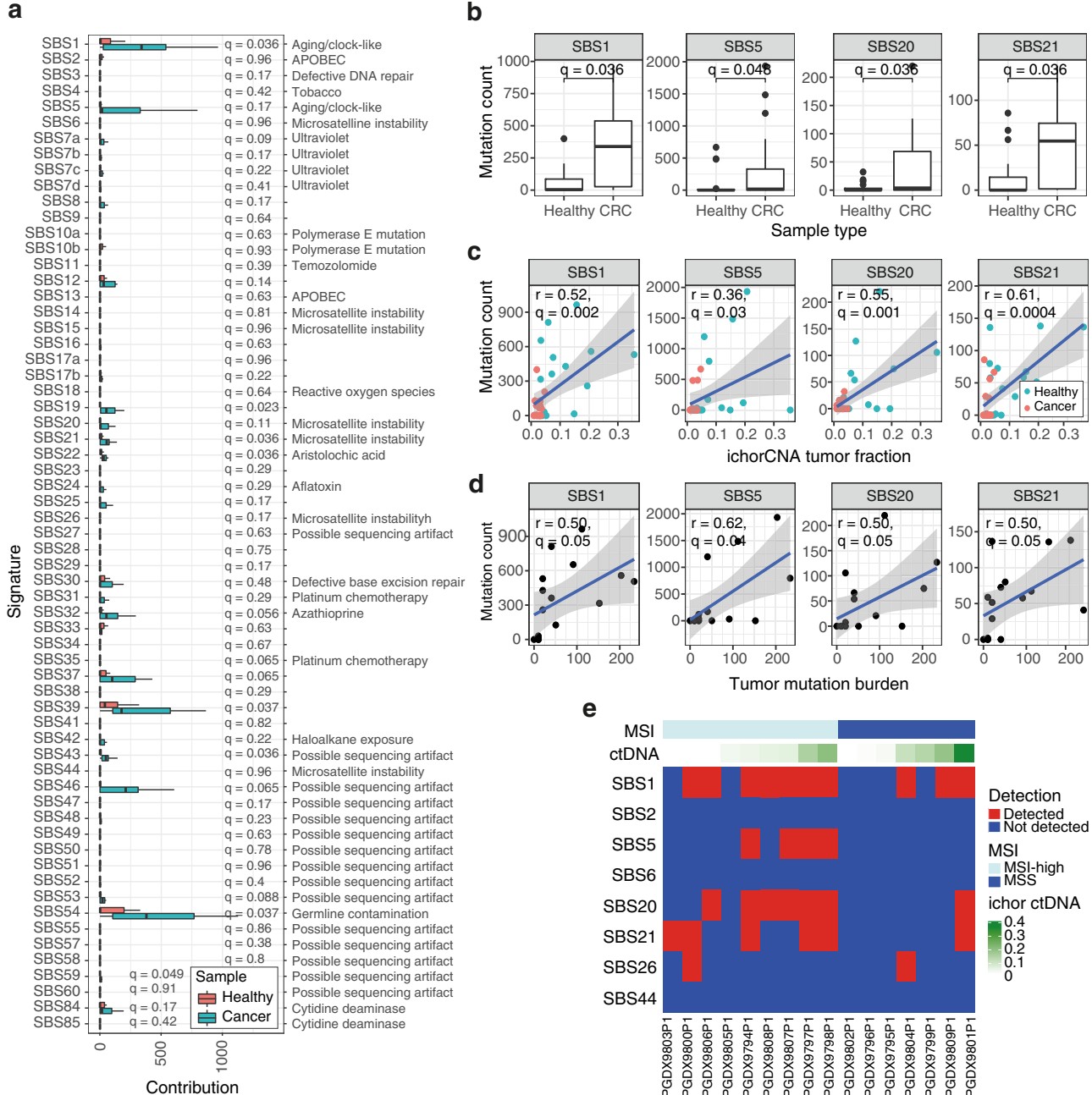

**Fig. 2 | Signature profiling in stage IV colorectal cancer (CRC). a** The number of mutations fitted to each signature is shown for plasma WGS data from healthy controls (*n* = 19, shown in red) and individuals with stage IV CRC (*n* = 16, shown in blue). Background subtraction was performed, and SNPs were retained ("Methods"). The potential etiology of each signature is listed[12,45]. Boxplots represent the median, upper and lower quartiles and whiskers indicate 1.5× IQR. One-sided Wilcoxon tests were performed, and Benjamini–Hochberg (BH)-corrected *p* values (*q*) are shown. APOBEC apolipoprotein B mRNA-editing enzyme. **b** Boxplots of aging and MSI signature contributions in healthy individuals (*n* = 19) and patients with CRC (*n* = 16). Mutation counts are background-subtracted ("Methods"). One-sided Wilcoxon tests were performed, and adjusted *p* values (*q*) are shown. Boxplots represent the median, upper and lower quartiles, and whiskers indicate 1.5× IQR. **c** One-sided Pearson correlations between tumor fraction and background-

subtracted mutation count for aging and MSI signatures are shown for healthy individuals (*n* = 19, shown in red) and patients with CRC (*n* = 16, shown in blue). Adjusted *p* values (*q*) are shown. The gray shaded area indicates the 95% CI of the fitted linear model. **d** One-sided Pearson correlations between tumor mutation burden (TMB) and mutation count per signature are shown for each patient (*n* = 16). Adjusted *p* values (*q*) are shown. The gray shaded area indicates the 95% CI of the fitted linear model. **e** Heatmap of selected aging and MSI signatures detected in CRC plasma samples with 95% specificity (*n* = 16, "Methods"). Detected signatures in each sample are indicated in red, non-detected signatures are shown in blue. Samples are annotated with ichorCNA ctDNA fraction and microsatellite instability status (MSI)[21]. Source data are provided as a Source data file. MSS microsatellite stable.

this approach across multiple cancer types. This cohort consisted of stage I–IV NSCLC (*n* = 37), stage I–III breast cancer (*n* = 48), stage I–IV CRC (*n* = 27), stage I–IV, 0 and X gastric cancer (*n* = 27), stages I, III, and IV ovarian cancer (*n* = 26), stage I–III pancreatic cancer (*n* = 34), and 206 individuals without cancer. Samples were analyzed across multiple

sequencers (Supplementary Fig. 12a), which showed batch differences in SBS profile in healthy individuals despite correction for GC-bias (Supplementary Fig. 12b). Therefore, cases were compared against controls from the same sequencer. Data were processed similarly to the previous cohort, with downsampling to 10M reads.

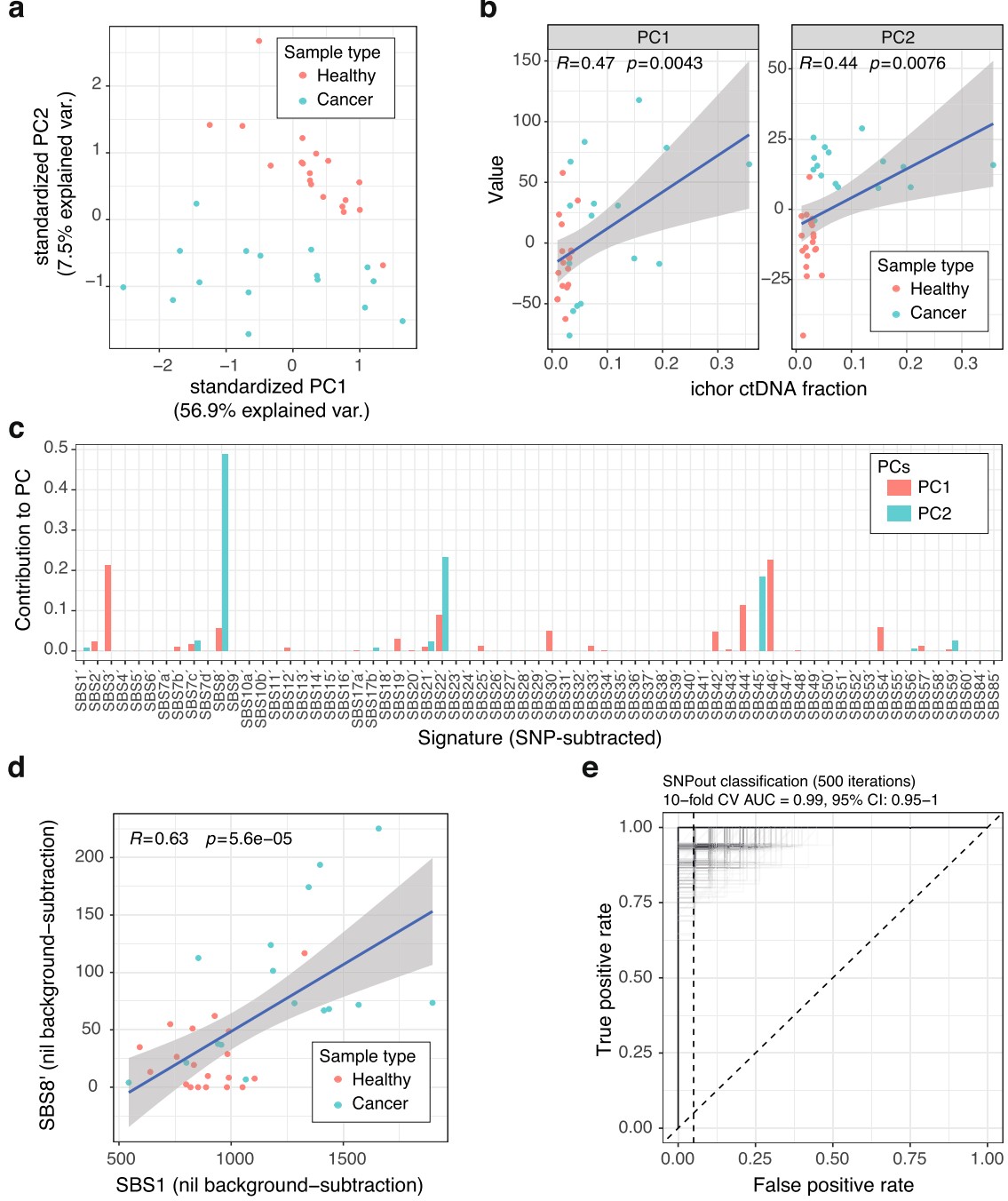

**Fig. 3 | Cancer detection in stage IV colorectal cancer (CRC). a** SNP-subtracted mutation profiles from 0.3× WGS of plasma from healthy individuals ($n = 19$) and patients with stage IV CRC ($n = 16$) were used as input for Principal Component Analysis (PCA). Healthy and cancer samples are shown in red and blue, respectively. PC principal component. **b** PC1 and PC2 were correlated against ctDNA fraction determined by ichorCNA. Both PCs showed significant correlation (PC1, $p = 0.0043$; PC2, $p = 0.0076$, two-sided Pearson correlation). Healthy and cancer samples are shown in red and blue, respectively. The gray shaded area indicates the 95% confidence interval of the fitted linear model. **c** The signature contributions to PC1 and PC2 were assessed by fitting signatures to the SBS profile of each PC. Signature contributions to each PC are shown as proportions. SBSn´ indicates SNP-subtracted mutation data fitted to SBSn, where $n$ is an integer. PC1 is shown in red, PC2 is shown in blue. **d** In samples from healthy individuals ($n = 19$) and patients with stage IV CRC ($n = 16$), SBS1 contribution was significantly correlated with SBS8´ ($p = 5.6 \times 10^{-5}$, two-sided Pearson correlation). Healthy and cancer samples are shown in red and blue, respectively. The gray shaded area indicates the 95% confidence interval of the fitted linear model. **e** A random forest model was used to classify cancer samples ($n = 16$) vs. healthy ($n = 19$) using SNP-subtracted mutation profiles. Ten-fold nested cross-validation repeated 500 times was used. Each iteration is shown. A Receiver Operating Characteristic curve is shown (AUC 0.99, 95% CI 0.95–1.00). Source data are provided as a Source data file. AUC area under the curve, CV cross-validation.

Signature fitting was performed with SNPs retained, as before, and all signature contributions are shown in Supplementary Data 4. Signatures previously identified in tumors by Alexandrov et al.[12] for each cancer type were used for signature detection with 95% specificity. Across the cohort, the proportion of patients with ≥1 signature detected ranged from 0.85 in NSCLC to 0.38 in pancreatic cancer (bootstrapped median, 100 iterations, Fig. 4a). By stage, the rate of detection of ≥1 signature ranged from 0.70 in stage I disease to 0.75 in stage IV disease (bootstrapped median, 100 iterations, Fig. 4b). Across all cancer types tested, none showed a significant correlation between

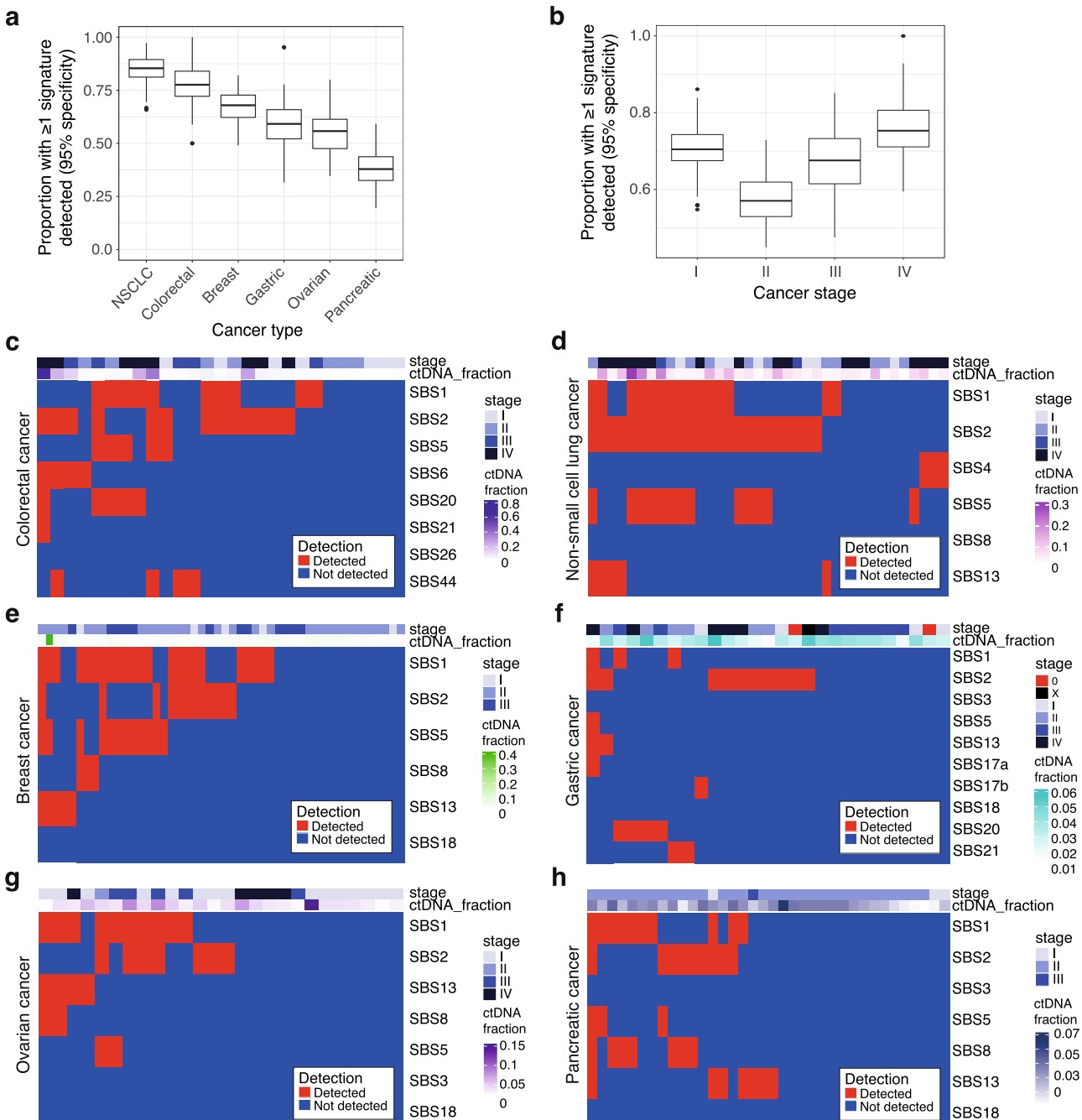

**Fig. 4 | Signature profiling of the DELFI cohort. a** Boxplots showing the proportion of patients from the DELFI cohort (*n* = 199) with at least one signature detected in plasma. Data are split by cancer type. Signatures relevant to each cancer type were assessed for detection with 95% specificity using 10M reads ("Methods"). Data were bootstrapped with 100 iterations. Boxplots show the bootstrapped median, upper and lower quartiles, and whiskers indicate 1.5× IQR. Outliers are shown as points. NSCLC non-small cell lung cancer. **b** Boxplots showing the proportion of patients with stage I–IV disease from the DELFI cohort (*n* = 197) with at least one signature detected in plasma are shown. Data are split by cancer stage.

Boxplots show the bootstrapped median, upper and lower quartiles, and whiskers indicate 1.5× IQR. Outliers are shown as points. **c–h** Heatmap of detected signatures for stage I–IV colorectal cancer (*n* = 43), stage I–IV NSCLC (*n* = 27), I–III breast cancer (*n* = 48), stage I–IV, 0, and X gastric cancer (*n* = 27), I–IV ovarian cancer (*n* = 26), stage I–III pancreatic cancer (*n* = 34). Detected signatures within a sample are shown in red, undetected signatures are shown in blue. Cancer type, disease stage, and ctDNA fraction are annotated. Source data are provided as a Source data file.

ichorCNA ctDNA fraction and the total number of plasma mutations following correction for multiple testing (Supplementary Fig. 12c). However, this correlation is limited by the ability to quantify ctDNA fractions in this cohort: only a median of 11.4% of samples was detected using ichorCNA with a 95% specificity threshold.

In patients with stage I–IV colorectal cancer, plasma signatures were detected in 21 out of 27 patients (78%, Fig. 4c). Aging signatures

were detected in 12 (44%), APOBEC signatures in 14 (51%), and MSI signatures in 11 out of 27 patients (41%). In stage I–IV NSCLC, signatures known to be associated with lung cancer[12] and tobacco exposure[18] were assessed. APOBEC signatures were the most prevalent signature detected (in 24 out of 37 samples, 65%), followed by aging (21 out of 37, 57%, Fig. 4d). SBS4 was observed in 3 out of 37 samples (8%), all of which were in stage IV disease. In patients with breast cancer, plasma

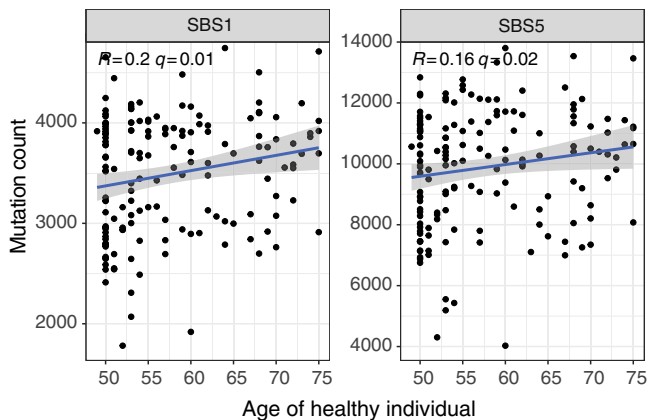

**Fig. 5 | Profiling aging signatures in healthy individuals.** One hundred and fifty-nine healthy individuals' plasma WGS data (50M reads) from the study in ref. 9 study, sequenced on the same machine, were used to explore the relationship between aging signatures in plasma and chronological age. We correlated known aging/clock-like[27] signatures (SBS1 and SBS5) against age. Both signatures showed a significant positive correlation with age (one-sided Pearson test). Adjusted *p* values (*q*) are shown. The gray shaded area indicates the 95% confidence interval of the fitted linear model. All putative aging-correlated signatures are correlated with age in Supplementary Figs. 13 and 14. Source data are provided as a Source data file.

signatures were detected in 67% (Fig. 4e); aging signatures were the most frequent (27 out of 48, 56%), followed by APOBEC signatures (16 out of 48, 33%). In patients with gastric cancer, APOBEC signatures were the most prevalent (11 out of 27, 41%), and evidence of MSI signatures was found in 22% (Fig. 4f). Two patients with stage 0 disease were included from the DELFI cohort, identifying an APOBEC signature in plasma in one case. In patients with stage I–IV ovarian cancer, aging signatures were the most frequent (9 out of 36, 35%, Fig. 4g). In patients with stage I–III pancreatic cancer, APOBEC signatures were the most frequently detected (11 out of 35, 31%), although overall detection rates were low (Fig. 4h).

The ratio between short (<150 bp) to long mutant fragments (>150 bp) was assessed as a quality control metric (Supplementary Methods). Patients with CRC, NSCLC, and breast cancer showed significantly shorter fragments than healthy controls (*q* < 0.005, two-sided Wilcoxon test, Supplementary Fig. 12d). These findings of short ctDNA fragments are consistent with the previous literature[8,25]. cfDNA from patients with pancreatic and gastric cancer were non-significantly longer in fragment size compared to healthy individuals (*q* = 0.53). There was a significant correlation between ichorCNA tumor fraction and short:long fragment ratio (Pearson *r* = 0.41, *p* = 6.9 × 10⁻⁸).

Given the high prevalence of SBS2 mutations detected with 95% specificity in ref. 9 sequencing data, we tested whether sequencing noise might contribute to SBS2 mutations. To quantify noise, we utilized the discordant mutations in the overlapping region of paired-end sequencing reads in each sample (Supplementary Fig. 12e). Discordance in mutations between overlapping R1 and R2 reads likely arise from sequencing noise[20], whereas true mutations would be present in both R1 and R2. The number of discordant mutations per sample was constant across each of the SBS contexts of SBS2 in patients with NSCLC compared to healthy individuals (*q* > 0.05, Supplementary Fig. 12f), suggesting that SBS2 calls are unlikely to arise from sequencing noise.

## Aging signatures in healthy individuals

Given the predominance of aging/clock-like signatures in Pointy data, we explored their relationship with chronological age in healthy individuals. Individuals with cancer were not used for this analysis to exclude tumor cells as a source of aging mutations. We

expected the magnitude of any relationship to be small based on previous estimates of aging mutation rates[18], combined with recent evidence for aging signatures varying between tissues[26]. One hundred and fifty-nine healthy individuals' plasma data arising from the same sequencer from ref. 9 study were used. Data were downsampled to 50M reads (1.5×) WGS, GC-normalized, and signatures fitted with SNPs retained. The age range of healthy individuals in this cohort was 49–75 years old, with a median age of 54 (Supplementary Data 5). Clock-like mutational signatures[27] (SBS1 and SBS5) were compared against chronological age using SNP-retained data. Both SBS1 and SBS5 showed a significant correlation with biological age (*p* < 0.022, one-sided Pearson correlation, Fig. 5).

To explore aging signals in other SBS signatures, signatures that were significantly correlated with SBS1 were identified as putative aging-correlated signatures (Supplementary Fig. 13a). These additional SBS1-correlated signatures were then compared against the chronological age of each healthy individual. Following correction for multiple testing, 13 further signatures showed a significant correlation with chronological age (*q* < 0.05, one-sided Pearson correlation, Supplementary Fig. 13b). With SNP-subtracted data, no mutations fitted to SBS1´ in this case due to bias introduced by SNP-subtraction (Supplementary Fig. 14a). Nonetheless, SBS2´, SBS30´, SBS33´, and SBS46´ mutation counts were significantly correlated with age (*q* < 0.02, Supplementary Fig. 14b). These data suggest that aging mutations may be detected in the plasma of healthy individuals, both in SBS1 and SBS1-correlated signatures, though the latter may be due to mis-attribution of aging mutations to other signatures due to signature similarity or biased fitting due to SNP removal.

## Cancer classification across cancer types

For all cancer types in the individual batch from the cohort in ref. 9 cancer detection and classification to cancer types were performed using SNP-subtracted SBS profiles. ichorCNA ctDNA fractions were included in each model, as before. Samples were downsampled to 25M (0.75×) reads and nested ten-fold cross-validation was used, repeated 500 times ("Methods"). For overall detection across all cancer types and stages (*n* = 199), an AUC of 0.96 was achieved (95% CI 0.94–0.98, Fig. 6). Across all stages, the AUC values were 0.99 for NSCLC (95% CI 0.97–1.00), 0.99 for breast cancer (95% CI 0.98–1.00), 0.98 for CRC (95% CI 0.96–0.99), 0.92 for gastric cancer (95% CI 0.84–0.98), 1.00 for ovarian cancer (95% CI 0.99–1.00), 0.88 for pancreatic cancer (95% CI 0.81–0.94, Supplementary Fig. 15a–f). Detection rates of patients across all stages was high (Supplementary Fig. 15g–j), as follows: stage I, AUC 0.96 (95% CI 0.90–1.00); stage II, AUC 0.95 (95% CI 0.92–0.98); stage III, AUC 0.97 (95% CI 0.93–0.99); stage IV, AUC 0.97 (95% CI 0.95–0.99). Detection rates by stage and cancer type with specificity set to 95% are shown in Supplementary Fig. 15k.

Based on differences observed in PC1 and PC2 between samples using PCA (Supplementary Fig. 15l), we assessed whether samples could be classified into individual cancer types. We selected patient samples sequenced on the same sequencer from the DELFI study (*n* = 70, Supplementary Methods). Healthy samples were excluded. Classification of individual cancer types achieved a median balanced accuracy of 0.80 (95% CI 0.52–0.89).

Lastly, we assessed the generalizability of this approach across cohorts, as patients with CRC were common to both cohorts (Supplementary Methods). We identified evidence of batch effect affecting SNP-subtracted mutation profiles of healthy controls between the two studies (Supplementary Fig. 16a), despite using quality filters and GC-bias correction. This may be due to differences in sample collection location, as cases were collected across multiple academic sites, with controls in the former study sourced from a commercial site[9,21]. To mitigate this, we pooled healthy and CRC patient samples across the

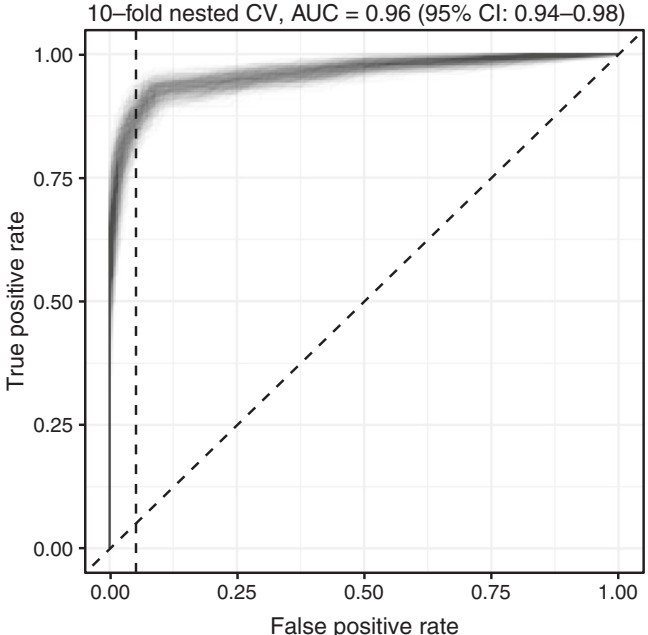

**10–fold nested CV, AUC = 0.96 (95% CI: 0.94–0.98)**

*y-axis:* True positive rate
*x-axis:* False positive rate

**Fig. 6 | Cancer detection across cancer types.** Receiver Operating Characteristic (ROC) curve for sample classification from the DELFI cohort, which included 199 individuals with cancer and 206 healthy individuals. A random forest model using ten-fold nested cross-validation with 500 iterations was used to classify samples as either healthy or cancer. This cohort consisted of stage I–IV NSCLC ($n = 27$); stage I–III breast cancer ($n = 48$); stage I–IV CRC ($n = 27$); stages I–IV, 0 and X gastric cancer ($n = 27$); stage I–IV ovarian cancer ($n = 26$); and stage I–III pancreatic cancer ($n = 34$). All ROC curves from 500 iterations are shown. Classification performance by individual cancer type and stage is shown in Supplementary Fig. 15. Source data are provided as a Source data file. AUC area under the curve; CI confidence interval; CV cross validation.

two studies in equal numbers to allow training across batches. Ten-fold nested CV was performed using RF, resulting in an AUC of 0.93 (95% CI 0.87–0.96, Supplementary Fig. 16b).

## Discussion

In this study, we identified mutational signatures in low-coverage plasma WGS in two independent datasets. Both exogenous and endogenous mutational processes were identified in plasma, including aging, smoking, APOBEC, and MSI signatures. We demonstrated sensitive cancer detection using these plasma signatures. Additionally, in healthy individuals, an age-correlated mutational signature was identified in plasma, consistent with previous findings in human tissues[11,13,18,28].

This study has notable limitations. We carried out this analysis with low-coverage WGS without matched germline samples, which improves the scalability of the approach but limits sensitivity for signature detection. This is particularly relevant for signature fitting, where the resolution for low-abundance and similar signatures was hampered by noise. Performing germline sequencing with low coverage sequencing alone would be of limited benefit: if 0.3× WGS (10M reads) were used to sequence a matched germline, only 13% of SNPs would be identified in the germline with 1 mutant read each. Germline calling conventionally requires >15× coverage[17,20], representing a 1–2 order of magnitude increase in sequencing, and would also need to be performed for healthy individuals to enable comparison. Future studies using deep sequencing in plasma and matched germline samples are needed to fully characterize circulating signatures.

To mitigate noise, we leveraged machine learning for the classification of samples and used quality filters ("Methods"). Differences in noise profile observed between control samples arising from

independent studies highlight the importance of validation of this approach on a larger scale across multiple cohorts. In the future, error-suppression using read collapsing may be employed, for example, by combining Pointy with laboratory methods to increase duplication rates[29]. Even with error-suppression, plasma signature profiles may still appear different from tumor signature profiles due to the effects of sampling error in plasma: low-frequency tumor mutational signatures may be missed, similar to the lower representation of heterogeneous tumor mutations in plasma[30].

Although subject to technical and biological noise, this proof-of-concept study of low-coverage plasma WGS provides an insight into circulating signatures in cfDNA and their potential utility in oncology. Deeper sequencing in future would enable the exploration of plasma signatures with greater detail. These signatures, whose exposures may be operative both before and during cancer development[10,13], might be used for earlier cancer detection. Sensitivity may be boosted further through utilizing ensemble machine learning approaches that combine mutational signatures with other parameters, such as cfDNA fragmentation patterns, fragment ends[31,32], and/or preferred end co-ordinates[33]. Lastly, deeper exploration of circulating mutational signatures in healthy individuals might enable the evaluation of cancer risk and the etiology of cancers.

## Methods

### Patient and sample characteristics

In this study, cfDNA WGS data were analyzed from a total of 215 patients and 227 healthy control individuals across two cohorts (Supplementary Data 1). For the initial cohort (PGDX), 16 patients with stage IV CRC provided plasma samples following written informed consent for research use as part of clinical trial NCT01876511. This protocol was approved by the Johns Hopkins Institutional Review Board[21,34]. Plasma samples from 21 healthy control individuals were procured through BioIVT[21].

We next studied 199 patients and 206 healthy control individuals from the DELFI[9] dataset following approval from their Data Access Committee (DAC). Samples in this cohort were obtained under Institutional Review Board approved protocols, with informed consent from all participants for research use at participating institutions[9]. Patients with the following cancer types were included from the DELFI study: CRC ($n = 27$), gastric ($n = 27$), NSCLC ($n = 37$), ovarian ($n = 26$), breast ($n = 48$), and pancreatic ($n = 34$). Only pre-treatment timepoints from the DELFI study were used. For this proof-of-concept study, no blinding or randomization was performed.

### Plasma sample preparation and sequencing

Plasma whole-genome library preparation was performed as described in refs. 21 and 9. Briefly, for both cohorts, cell-free DNA (cfDNA) was extracted from plasma using the QIAamp Circulating Nucleic Acid Kit. Libraries were prepared with 5–250 ng of cfDNA using the NEBNext DNA Library Prep Kit. Whole-genome libraries were sequenced using 100 bp paired-end runs on HiSeq 2000/2500 sequencers (Illumina).

### Whole-genome sequencing data processing

An overview of the pipeline used is shown in Supplementary Fig. 1. Raw FASTQ files were trimmed using trimmomatic (version 0.39)[35] in paired-end mode, with the following settings: all reads were cropped to 100 bp (CROP: 100), Illumina sequencing adaptors were removed (ILLUMINACLIP: 2:30:10:2:keepBothReads), leading and trailing 3 bp were trimmed if they were low quality (LEADING: 3, TRAILING: 3), and reads with an average base quality <30 were removed (AVGQUAL: 30). For public datasets, where BAM files were provided, we converted each BAM file to FASTQ using Bedtools (version 2.28.0) bamtofastq prior to running trimmomatic.

Trimmed FASTQ files were aligned to the hg38 genome using BWA (version 0.7.15) mem, sorted and indexed with samtools (version 1.7), and duplicates marked and removed with Picard (version 2.19.0) MarkDuplicates. Indel realignment was performed with GATK (version 3.8). Each BAM was downsampled using Picard (version 2.19.0) DownsampleSam to either 10M reads (PGDX cohort - signature profiling and classification; DELFI cohort - signature profiling), 25 M reads (DELFI cohort - classification), or 50M reads (DELFI cohort - signatures in healthy individuals, Supplementary Data 1). BAM files with <90% of the target number of reads for downsampling were excluded. To maximize the quality of the mapped reads, downsampled BAMs were intersected with UCSC tracks WindowMasker[36] and RepeatMasker to remove repeats, then were intersected to retain only regions in the GATK WGS calling regions BED from the GATK hg38 resource bundle. Reads with secondary mapping positions were removed with grep. Reads with a fragment length of zero were removed with awk, as were reads with any supplementary alignments.

Each BAM file was converted to SAM using samtools (version 1.7) and then filtered using awk to retain mutant reads containing a single point mutation only. Samtools mpileup (version 1.7) was used to identify point mutations, considering only reads with a mapping quality of 60 (-q) and considering mutations only if they had a minimum base quality of 30 (-Q). Indels were removed from the mutation VCF using grep. ANNOVAR (version 2018-04-16) was used to annotate variants using RefSeq[37] and dbSNP 151[38]. Mutations were annotated as being either concordant, i.e., supported by both R1 and R2 of the same mate pair, or discordant. Annotated and filtered VCFs were read into R (version 4.1.2) and mutations were annotated with single base substitution contexts using the MutationalPatterns package (version 1.10.0)[39] and processed with dplyr (version 1.0.8) and plyr (version 1.8.7).

For all samples, the sequencer ID was obtained from the read header in the FASTQ file using a custom shell script. To minimize sequencer-specific batch effects on signature profile analysis and sample classification, all downstream analyses were batched by sequencer, with patient samples being controlled by healthy individuals on the same sequencer. Two sequencer IDs were excluded due to few samples or only healthy samples being present (Supplementary Fig. 12a).

## GC normalization

A GC-normalization step was performed to correct for differences in GC profile between samples. First, a GC profile was first determined for each downsampled BAM file. GC bias metrics were generated using Picard (version 2.19.0) CollectGcBiasMetrics with a WINDOW_SIZE of 300 bp based on previous literature on GC bias in cfDNA[40]. Next, GC profiles for all samples belonging to the same sequencer ID were loaded into R, and a generalized additive model (GAM) was used to generate an averaged profile for the batch, using ggplot2 (version 3.3.5) geom_smooth() using method = "gam" and formula = "y ~ s(x, bs = "cs")." Supplementary Fig. 3c shows GC profiles for samples sequenced on two runs from the same sequencer, plus the GAM averaged GC profile.

The averaged GC profile was used to normalize the mutation counts of all samples, based on the GC content of each mutated read, as follows: a custom R script was used to annotate all mutations in each sample with their associated GC sequence content, rounded to the nearest 1%. The number of mutations in each GC content % bin was normalized relative to the averaged GC profile of that batch, aiming to mitigate differences in GC bias.

## Mutational signature profiling and detection in patient samples

For analysis of mutational signatures in patient plasma samples in both cohorts, a 96-SBS mutation profile was generated for each sample, as described above. For each of the 96 SBS contexts in all samples (cases and controls), the median number of background mutations in that SBS context in control samples was subtracted. Background subtraction was performed relative to control samples sequenced on the same sequencer.

Mutational signatures were fitted to 96-SBS mutation profiles using the fit_to_signature() function from MutationalPatterns (version 1.10.0)[39]; SBS signatures published by Alexandrov et al.[12] were used as the reference signature matrix. This function finds "the optimal non-negative linear combination of mutation signatures to reconstruct the mutation matrix"[39]. Mutations that had been annotated as SNPs were retained for analysis of signature profiles, as we showed that removal of SNPs on these data can distort signature fitting processes due to high contributions of aging mutations among SNPs (Supplementary Fig. 10). After signature fitting, a matrix of signature contributions for each sample was generated.

To determine whether the signature contribution in an individual sample was significantly above background, we set 95% specificity thresholds for signature detection/calling based on values in control samples. Bootstrapping with 100 iterations in controls was used to generate each 95% specificity threshold.

## Mutational signature profiling in healthy individuals

For signature profiling in healthy individuals from the DELFI study, all healthy individuals sequenced on the sequencer named HISEQ were analyzed (n = 159). Signature fitting was performed as above, except background subtraction was not performed (as all samples were controls). Signature contributions were correlated against healthy individuals' chronological age from DELFI metadata and visualized using ggplot2 (version 3.3.5) and ggpubr (version 0.4.0).

## Sample classification

For sample classification, SNPs were subtracted to maximize signal:noise. 96-SBS mutation matrices were used as input. For all samples, PCA was used to reduce dimensionality using the stats package in R, and Principal Components with <1% variability were removed as a feature selection step. For each sample, a matrix of PCs, annotated with ichorCNA ctDNA fraction, was used as input for the classification model. Samples were classified using controls from the same study and from the same sequencer. For sample classification to either healthy or cancer, we tested multiple classification methods using a nested ten-fold cross-validation method[41], repeated 10 times, using: xgboost, Random Forest (RF), Support Vector Machine (SVM) and Logistic Regression. Nested k-fold cross-validation develops a new model on each training set, with testing on the held-out fold, and has been suggested to be robust to limited sample size[41,42]. CreateFolds() from the caret package (version 6.0–90) was used to generate balanced folds for each round of cross-validation.

xgboost (version 1.5.2.1) was used in R with the default parameters and nrounds = 100. randomForest (version 4.6–14) was used in R with the default parameters and ntree = 500. For SVM, svm() from the e1071 package (version 1.7.9) was used with default settings. For logistic regression, glm() from the stats package (version 4.1.2) was used with default settings. Following each iteration of cross-validation, a Pointy score for each sample was generated, ranging from 0 to 1 (higher represents more likely to be cancer). Classification performance characteristics were determined using the ci.cvAUC function from the cvAUC package (version 1.1.0) in R, using Pointy scores from all iterations as input. Random Forest showed the highest performance (Supplementary Fig. 11) and was selected for use for sample classification with nested 10-fold cross-validation with 500 iterations. Median Pointy scores are shown in Supplementary Data 1. Pointy scores from all iterations from all samples from each study were used as input into ci.cvAUC() to generate AUC values for each iteration by cancer type and stage (Fig. 3e and Supplementary Fig. 15).

## Classification of cancer type

For classification of samples to individual cancer types, cancer samples from the DELFI cohort one sequencer was used (HWI-D00837). Plasma WGS data were downsampled to 25M reads. For each sample, PCs were extracted from the 96-SBS mutation matrix belonging to each sample (as before), and these were used as input into a random forest classifier. Samples were classified to any of the cancer types present in the dataset using nested ten-fold cross-validation, repeated 10 times. This classifier generates a probability of matching the sample to each class (i.e., cancer type), and the highest scoring class was chosen as the predicted class. In the event of ties between classes, these were resolved using ties.method = "last". The classification performance was assessed using a confusion matrix from the caret package (version 6.0–90) in R.

## ctDNA fraction quantification using ichorCNA

For all plasma and tumor samples, the ctDNA fraction (termed as the tumor.fraction) was calculated using ichorCNA (version 0.2.0)[5], using a window size of 1 mb (--window), minimum quality of 20 (--quality), across all autosomes and sex chromosomes (--chromosome), with a maximum copy number of 3 (--maxCN). A panel of normals was not used; instead, ichorCNA was run across all healthy control samples within each batch. Detection thresholds for ichorCNA were determined in the DELFI cohort using a 95% specificity threshold of ctDNA fractions in healthy individuals in that cohort.

## Fragmentation analysis

To analyze the fragment size of Pointy mutations, insert sizes were obtained from the SAM file belonging to each sample. Each raw mutation matrix containing concordant mutations (i.e., present in both F and R mate pairs) was annotated with the insert sizes from the SAM file using a custom R script. Fragments with an insert size >1000 bp were excluded. A short:long fragment size ratio was calculated for each sample using a threshold of 150 bp for short fragments.

## Reporting summary

Further information on research design is available in the Nature Research Reporting Summary linked to this article.

# Data availability

Plasma whole-genome sequencing data have been deposited at the European Genome-phenome Archive (EGA), which is hosted by the EBI and the CRG, under accession number EGAS00001006377. Sequence data from the Cristiano et al.[9] study were previously deposited at the EGA, under accession number EGAD00001005339. Further information about EGA can be found on https://ega-archive.org "The European Genome-phenome Archive of human data consented for biomedical research"[43]. The sequencing data are available under restricted access to comply with patient consent for data sharing, access can be obtained by approval via their respective Data Access Committees via the EGA. The following public databases were used to annotate mutations: 1000 Genomes[23], RefSeq[37], and dbSNP 151[38]. Source data are provided with this paper.

# Code availability

Code used in the Pointy pipeline is available for academic research purposes only[44] at https://doi.org/10.5281/zenodo.6666951. Code is in a restricted-access repository; users are required to agree to the license terms and conditions prior to approval. We aim to respond to data access requests within 5 working days.

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

## Acknowledgements

The authors would like to thank Jillian Phallen and Victor Velculescu for their assistance with ref. 9 dataset. This research is supported by an NIH Core Grant (P30 CA008748) and a Stand Up to Cancer Colorectal Cancer Dream Team Translational Cancer Research Grant (SU2C-AACR-DT22-17). Stand Up to Cancer is a program of the Entertainment Industry Foundation administered by the American Association for Cancer Research. B.R. is supported by a Swim Across America Foundation Fellowship, the Nuevo Soldati Foundation, and the Dalton Family Foundation.

## Author contributions

J.C.M.W. and L.A.D. conceptualized the study. J.C.M.W., D.S., L.L., and J.R.W. performed the analyses. C.S. and B.R. performed experiments. D.W.Y.T. and L.A.D. supervised the project. J.C.M.W. and L.A.D. wrote the manuscript, and all authors approved the final draft.

## Competing interests

J.C.M.W. is an inventor of patents for methods for ctDNA detection. J.C.M.W. and L.A.D. filed provisional patent application "Detection of somatic mutational signatures from whole genome sequencing of cell-free DNA" (U.S. Provisional Patent Application Number 63/216,727) relating to this work. D.S. serves as a consultant for Pyxis, Jounce, and Memorial Sloan Kettering Cancer Center and is employed by the Dana-Farber Cancer Institute. J.R.W. is the founder and owner of Resphera Biosciences LLC, and serves as a consultant to MSKCC and Personal Genome Diagnostics Inc. B.R. has served in a consulting/advisory role for Bayer, Roche, Novartis, Gilead, Servier, and Neophore and has received travel expenses from Bayer, Servier, and Astellas. D.W.Y.T. received the following honoraria more than two years ago from Nanodigmbio, Cowen, and BoA Merrill Lynch; research funding from ThermoFisher Scientific, EPIC Sciences, Prostate Cancer Foundation; Travel, Accommodation, Expenses from Nanodigmbio. D.W.Y.T. is a co-inventor of "Systems and methods for detecting cancer via cfDNA screening" (WO2019204208A1) and "Systems and methods for distinguishing pathological mutations from clonal hematopoietic mutations in plasma cell-free DNA by fragment size analysis" (WO2021194837A1). D.W.Y.T. is currently an employee of PetDx. Inc. L.A.D. is a member of the board of directors of Personal Genome Diagnostics (PGDx) and Jounce Therapeutics. He is a compensated consultant to PGDx, 4Paws (PetDx), Innovatus CP, Se'er, Kinnate, and Neophore. He is an uncompensated consultant for Merck but has received research support for clinical trials from Merck. L.A.D. is an inventor of multiple licensed patents related to technology for circulating tumor DNA analyses and mismatch repair deficiency for diagnosis and therapy from Johns Hopkins University. Some of these licenses and relationships are associated with equity or royalty payments directly to Johns Hopkins and L.A.D. He holds equity in PGDx, Jounce Therapeutics, Thrive Earlier Detection, Se'er, Kinnate, and Neophore. His spouse holds equity in Amgen. The terms of all these arrangements are being managed by Johns Hopkins and Memorial Sloan Kettering in accordance with their conflict of interest policies. L.L. and C.M.S. declare no competing interests.
