## [Peer review file · Nature Communications]

Editorial Note: Parts of this Peer Review File have been redacted as indicated to maintain confidentiality of private information.

REVIEWER COMMENTS

Reviewer #1 (Remarks to the Author):

The authors have replied additional sentences for the reviewer 1's suggestions. The reviewer 1 accept these sentences.

Reviewer #3 (Remarks to the Author):

The authors have addressed the issues raised by me and also by the other reviewers and I think the paper is now acceptable

Reviewer #4 (Remarks to the Author): Expert in computational genomics, bioinformatics, and liquid biopsies

NCOMMS-21-37107-T

Wan and colleagues present a study on mutational signature analysis from low-coverage genome sequencing data of circulating tumor DNA (ctDNA). They describe an approach called "Pointy" which analyzes mutational signatures in 0.3-1.5x depth sequencing data. The authors describe some technical challenges and over-coming them, such as using GC-correction, background-subtraction, and use of SNP-subtraction. Signature and its contributions were estimated by signature fitting using MutationalPatterns package. The first application of Pointy is for the analysis of mutational signatures in Stage IV colorectal cancer (CRC) from patient plasma and healthy donors. Benchmarking analysis was performed using in silico simulations. Then, they focused on describing aging and MSI signatures in CRC, followed by cancer detection in this discovery cohort. The remaining analyses was performed on a pan-cancer plasma dataset published by Cristiano et al. The authors describe the contribution of APOBEC

and smoking signatures in various cancer types, and then describe aging signatures in the healthy donors.

Finally, the authors demonstrate cancer detection and cancer type classification on this cohort.

This study is very interesting to the liquid biopsies research field and has the potential advantage of being a cost-efficient approach to analyze mutational signatures. This is a proof-of-concept approach that addresses technically more challenging type of data but is still based on the huge body of literature that supports the importance of mutational signature analysis. There are several important analysis related issues I have with this study that limits my enthusiasm for its potential applications and ultimately its utility. My concerns are the lack of rigor or clarity in the signature simulation benchmarking; problems of overfitting for the CRC MSI classification and cancer detection; lack of independent validation cohort for trained models; lack of confirmation of cancer-related signatures; and overall low quality/detail in the description of the methods is hard to follow and may make reproducibility difficult.

1. Modeling of mutant reads in low-coverage sequencing data does not account for non-uniformity of coverage due to cfDNA fragmentation and nucleosome occupancy. While this analysis is supposed to motivate the need for overcoming technical challenges of low coverage, it does not add much to the manuscript. It is too simplistic and not representative of cfDNA. Furthermore, the authors have not learned any meaningful statistics from this to actually inform their Pointy method. For example, what is the theoretical lower limit of ctDNA fraction to detect a single mutation signature at a given seq coverage at some statistical power; or the theoretical seq coverage limit at a given ctDNA fraction at some power to detect a single signature; or the theoretical limit of required number of mutations at a given ctDNA fraction and seq coverage at some power to detect a single signature. I think this section can be relegated.

a. How does the error suppression improve on the analysis? It is important to know this, especially if no germline controls are included. Please provide an evaluation on the benefits of error-suppression in your analysis.

2. GC-correction.

a. The authors should provide a figure showing the SBS profiles as a function of GC content, 1 panel for uncorrected and 1 panel for GC-corrected, to directly highlight/emphasize the extent of the GC bias for each sequencing run. Can be an inclusion in Extended Fig 3.

b. This figure can also clarify and make clear what is being fit using the generalized additive model and how it is leading to a new corrected SBS profile. The methods section for this part is a bit abstract and more details should be included to make it more clear.

3. Fragment size analysis of mutant reads (Lines 133-139). Did the analysis that revealed a 2bp shorter length in cancer samples include all variants, both SNVs and SNPs?

a. The ~148bp fragment length for mutation reported sticks out as very discrepant from the expected 167bp fragment length of cfDNA. While this is likely because Pointy requires both R1 and R2 reads contain the mutation, it can be misleading if the reader is not aware of this detail. I would question and weigh whether this analysis is providing anything of value in light of its potential to be misleading.

4. In silico mutation fitting and performance evaluation. This is an important and necessary inclusion in the manuscript, so it is great to see this systematic evaluation. However, the methods section in the supplement is far too abstracted and details are simply glossed over, such that it is not easy to interpret or likely reproduce the results.

a. How were the 10-1000 mutations selected? What is the source?

b. The definition of efficiency needs to be properly defined in the methods text—currently it is found in the captions of Extended Data Fig 5 and 6. What is the “expected increase” in signature in the efficiency definition and how is this defined? What is the ground truth signature contribution after spike-in and how is that defined?

c. Is the evaluation based on assessing the “contribution of signatures” or the “contribution of mutations to a given signature”? This is not clear. The contribution of signatures seems to be important but it’s not clear how the ground truth (expected contribution) is computed.

d. Was only a single signature spiked-in? What is the ability and performance of Pointy for estimating multiple signatures within a sample, which can often be the case in cancer.

e. Show example SBS profiles for spike-in’s of 10, 100, 1000 to emphasize the difference in the input data signals for the number of mutations.

f. Extended Data Fig 5 caption lists Pearson $r=1 \times 10^{-12}$ instead of Pearson $r=0.71$, $p=1.6 \times 10^{-11}$.

g. Main text says “94% for 500 mutations but in Extended Data Fig 5a, it shows 10, 100, 1000 mutations.

5. Signature calling thresholds. What in silico experiment does Line 187 refer to? How did the authors arrive at 13.9 median number of mutations (Extended Data Fig. 8) required to call a signature come? Please describe in the methods section.

6. Classification of MSI vs MSS. The analysis using signature SBS20 to classify MSI CRC sample is not very convincing both for the low sample size, lack of validation cohort, technical issues with the machine learning procedure, and that there are likely other features more predictive of MSI.

a. According to the methods, 10-fold cross-validation was used. Since only 7 MSS samples are available, even with stratified CV, some of the folds will not contain MSS samples. Likely this will result in over-fitting anyway because of the small sample size. The authors may want to consider leave-one-out CV (jack-knifing).

b. 16 CRC samples (11 MSI vs 5 MSS) is small and fine as a discovery cohort for this proof-of-concept. However, since this cohort has such small sample size, an independent validation cohort to test the trained model is required. The trained model on the 16 samples should be used to predict MSI status in a different test set cohort.

c. The usefulness of this model is trumped by simply using TMB. Aside from PGDX9805P1, all other MSI-H samples have high TMB.

7. Cancer-relevant mutational signature prediction shows some interesting signals in the appropriate cancer types. It would be a stronger result if further confirmation came from deeper coverage sequencing where mutations can be called more accurately. For example, analyze signatures in a publicly available dataset of deeper whole genome sequencing and apply Pointy to the down-sampled version.

8. Cancer detection analysis in CRC cohort.

a. Line 246 refers to xgboost and directs the reader to the Methods but it is not described in the Methods at all.

b. The small sample size of 16 CRC vs 20 healthy may suffer from overfitting, since standard 10-fold CV and only 50 iterations is performed. To properly evaluate the performance, the authors should use the trained model and predict on a different test set cohort. The Cristiano et al. DELFI CRC dataset would be ideal.

9. Analysis of mutant fragments in DELFI cohort.

a. Why is breast cancer not included in the Extended Data Fig 12d?

b. Is the observation in Extended Data Fig 12d partially explained by ctDNA fraction? CRC appear to have the highest tumor fractions and gastric & pancreatic likely have lower tumor fraction.

10. Cancer detection analysis in DELFI cohort is not really considered an independent validation cohort for the purposes of the machine learning classification.

a. As mentioned in Comment #8b, a proper use of an independent validation cohort is to apply the trained model from the discovery cohort to this validation cohort.

b. For reporting cross-validation performance on the DELFI cohort is fine but model overfitting may still be a limitation, especially when only 50 iterations were performed. Thus, the AUC values are not likely sampled enough to represent the true distribution of values. If bootstrapping is to be used, then it should follow the procedure of drawing samples into the training set with replacement, which will lead to ~63% of samples for training. Furthermore, since the model training was carried out on this same dataset, performance should be based on evaluation of an independent dataset (which was not part of the cross-validation). That is, use the trained model to predict cancer in a different test set.

c. How were the 137 cancer patient samples selected when there are 208 available? How were the 47 healthy donors selected when there are 215 available? Please describe in the Main text and Method section for the rationale to use a subset of samples with the same sequencing machine ID. How much of an impact does different sequencing machine for this dataset have on this approach? This seems to be quite important for your analysis, but no justification was provided. If GC-correction is already part of the approach to adjust batch effects, then this appears to be another potential artifact that has not been clearly shown for this datatype and its impact on the approach. Please provide data to support this.

11. The link to the code does not work or files have been removed.

I was also asked to evaluate the response to the concerns that Reviewer #2 raised. Please see the attached file for my comments on which concerns still need to be addressed.

Reviewer #1 (Remarks to the Author):

The authors have replied additional sentences for the reviewer 1's suggestions. The reviewer 1 accept these sentences.

Reviewer #3 (Remarks to the Author):

The authors have addressed the issues raised by me and also by the other reviewers and I think the paper is now acceptable.

Reviewer #4 (Remarks to the Author): Expert in computational genomics, bioinformatics, and liquid biopsies

Wan and colleagues present a study on mutational signature analysis from low-coverage genome sequencing data of circulating tumor DNA (ctDNA). They describe an approach called “Pointy” which analyzes mutational signatures in 0.3-1.5x depth sequencing data. The authors describe some technical challenges and over-coming them, such as using GC-correction, background-subtraction, and use of SNP-subtraction. Signature and its contributions were estimated by signature fitting using MutationalPatterns package. The first application of Pointy is for the analysis of mutational signatures in Stage IV colorectal cancer (CRC) from patient plasma and healthy donors. Benchmarking analysis was performed using in silico simulations. Then, they focused on describing aging and MSI signatures in CRC, followed by cancer detection in this discovery cohort. The remaining analyses were performed on a pan-cancer plasma dataset published by Cristiano et al. The authors describe the contribution of APOBEC and smoking signatures in various cancer types, and then describe aging signatures in the healthy donors. Finally, the authors demonstrate cancer detection and cancer type classification on this cohort.

This study is very interesting to the liquid biopsies research field and has the potential advantage of being a cost-efficient approach to analyze mutational signatures. This is a proof-of-concept approach that addresses technically more challenging type of data but is still based on the huge body of literature that supports the importance of mutational signature analysis. There are several important analysis related issues I have with this study that limits my enthusiasm for its potential applications and ultimately its utility. My concerns are the lack of rigor or clarity in the signature simulation benchmarking; problems of overfitting for the CRC MSI classification and cancer detection; lack of independent validation cohort for trained models; lack of confirmation of cancer-related signatures; and overall low quality/detail in the description of the methods is hard to follow and may make reproducibility difficult.

4.1 Many thanks for your interest in our work and for taking the time to provide detailed feedback. We have sought to address each of your queries in the responses below.

Notably, through expanding the cohort size to $n = 442$ (215 patients, 227 healthy) following your feedback, we now show:

1) Signature detection in plasma ranged from 84% in stage I-IV NSCLC to 40% in stage I-III pancreatic cancer (Fig. 4a) and ranged from a median of 61% detection in stage I-III disease to 78% in stage IV disease (Fig. 4b). Individual signature calls are shown in Figs. 4c-h.

2) We are now better powered to detect physiological signatures in healthy individuals. We now detect a significant correlation between SBS1 in plasma and chronological in healthy individuals (Fig. 5b), suggesting the possibility of pre-cancer signature profiling in future.

3) We confirm an AUC of 0.94 for cancer detection using 10-fold nested cross-validation (repeated 500 times) in the expanded DELFI cohort across cancer types (Fig. 6). Furthermore, we now combine both PGDX and DELFI cohorts' data to show an AUC of 0.84 for detection of CRC (Extended Data Fig. 16). Please see Response 4.17 for discussion on cross-validation on this dataset.

1. Modeling of mutant reads in low-coverage sequencing data does not account for non-uniformity

of coverage due to cfDNA fragmentation and nucleosome occupancy. While this analysis is supposed to motivate the need for overcoming technical challenges of low coverage, it does not add much to the manuscript. It is too simplistic and not representative of cfDNA. Furthermore, the authors have not learned any meaningful statistics from this to actually inform their Pointy method. For example, what is the theoretical lower limit of ctDNA fraction to detect a single mutation signature at a given seq coverage at some statistical power; or the theoretical seq coverage limit at a given ctDNA fraction at some power to detect a single signature; or the theoretical limit of required number of mutations at a given ctDNA fraction and seq coverage at some power to detect a single signature. I think this section can be relegated.

4.2 Thank you for your feedback on this figure. We have now removed it from the manuscript, and we now highlight the technical challenges associated with low depth of sequencing in the introduction instead (Line 63 – N.B. line numbers relate to ‘Simple Mark-up’ view).

a. How does the error suppression improve on the analysis? It is important to know this, especially if no germline controls are included. Please provide an evaluation on the benefits of error-suppression in your analysis.

4.3 We have now performed a comparison of error-suppressed vs. non-error-suppressed data as input for cancer detection using a random forest model in the PGDX cohort, and have updated the manuscript, as follows:

Line 259 - We also tested the effect of error-suppression, i.e. requiring mutations to be supported in both F and R mate pairs vs. being supported in either F or R only, which showed AUC values of 0.98 vs. 0.93 (P = 0.004, Wilcoxon test, Extended Data Fig. 11h). Therefore, for subsequent analyses for cancer detection, we processed data by using (a) SNP-subtraction, (b) PC transformation, (c) mutations only in both F and R reads, and (d) detection using an RF model (Methods).

Extended Data Fig. 11. Performance comparison of data processing steps and machine learning models. ... (h) Classification of samples using mutations supported by both F and R read of each paired-end read (error-suppressed) vs. either F or R read (non-error-suppressed) showed significant benefit of error-suppression (AUC 0.98 vs. 0.93, P = 0.004, Wilcoxon test).

2. GC-correction.

a. The authors should provide a figure showing the SBS profiles as a function of GC content, 1 panel for uncorrected and 1 panel for GC-corrected, to directly highlight/emphasize the extent of the GC bias for each sequencing run. Can be an inclusion in Extended Fig 4.

4.4 Certainly, we have now added additional panels (g) and (h) to Extended Data Fig. 3, where we analyze the similarity in SBS profile between healthy samples from the PGDX cohort that were sequenced on the same sequencer in two separate runs, with and without GC-correction. In Extended Data Fig. 3g, we show high cosine similarities between sample even without GC-correction, though this increased significantly following GC-correction (0.995 vs 0.999, $P < 2.2 \times 10^{-16}$, Wilcoxon test). Given the high degree of similarity between batches, rather than showing the individual SBS profiles, the differences between SBS profiles with and without GC-correction are shown in Extended Data Fig. 3h.

Extended Data Fig. 3. GC normalization. (a) Stage IV colorectal cancer samples from the PGDX cohort were sequenced in two batches on the same sequencer, enabling the study of inter-batch differences. There was no significant difference between the total number of mutations per sample

between batches ($p = 0.48$, two-sided Wilcoxon test). Box plots represent median, bottom and upper quartiles, and the whiskers correspond to $1.5\times$ interquartile range (IQR). **(b)** However, PCA of SBS profiles of the same samples showed clustering by sequencing run. **(c)** The GC content per bin is shown for samples, colored by batch. The average GC content profile across both batches is shown with a black dotted line, against which all samples are normalized (Methods). **(d)** Before GC-correction, there was a significant difference between sequencing runs in PC2 without p-value correction. Correction for multiple testing was not used to maximize sensitivity for possible batch effects. **(e)** The SBS profiles of PC1 and PC2 are shown, indicating that PC2 is primarily composed of SBS contexts at the extremes of GC-content. **(f)** Following GC-bias correction, there was no significant difference in any PC. This GC-correction step was therefore incorporated into the pipeline. **(g)** The cosine similarity in SBS profile between healthy samples from each batch (118 vs. 119) was compared with and without GC-correction, using bootstrapping with 100 iterations. GC-corrected samples showed significantly greater cosine similarity (0.999 vs. 0.995, $P < 2.2 \times 10^{-16}$, Wilcoxon test). **(h)** The difference in SBS profiles between each batch (118 vs. 119) is shown for uncorrected (upper) and GC-corrected data (lower). GC correction reduces the magnitude of GC-bias.

Line 101 -

The samples from the PGDX cohort were sequenced in two batches from the same sequencing instrument, so we explored data from healthy individuals for batch effects. In healthy samples, there was no significant difference in the mean number of mutations between batches (9,049 vs. 10,089, $p = 0.47$, two-tailed Wilcoxon test, Extended Data Fig. 3a). However, Principal Component Analysis (PCA) of SBS profiles revealed evidence of batch effect difference in mutation profile (Extended Data Fig. 3b), which may arise from differences in GC-bias between sequencing runs (Extended Data Fig. 3c). We identify a small but significant difference in the mean contribution of PC2 per sample (unadjusted $p = 0.022$, two-tailed Wilcoxon test, Extended Data Fig. 3d). The largest contributors to PC2 were contexts at the extremes of GC content (Extended Data Fig. 3e). Therefore, the GC bias for each sample was determined, as was the average GC profile of the sequencing batch, which were combined to normalize the SBS profile of each sample (Methods). This approach is analogous to GC-correction methods used to correct whole genome copy-number^{11,22} or fragmentation profiles¹⁵. After GC-correction, there were no significant differences in any PC between the two sequencing runs (unadjusted $p > 0.05$, two-tailed Wilcoxon test, Extended Data Fig. 3f). In Extended Data Fig. 3g, we show high cosine similarities between sample even without GC-correction, though this increased significantly following GC-correction (0.995 vs 0.999, $P < 2.2 \times 10^{-16}$, Wilcoxon test). The difference in SBS profiles between batches with and without GC-correction is shown in Extended Data Fig. 3h.

b. This figure can also clarify and make clear what is being fit using the generalized additive model and how it is leading to a new corrected SBS profile. The methods section for this part is a bit abstract and more details should be included to make it more clear.

4.5 Of course, we have now clarified the steps involved in for the generalized additive model fit and GC-correction, please see below. We have also added Extended Data Fig. 3c to show the coverage per GC bin (rounded to 1%) for samples sequenced on two separate machines, plus the averaged fitted profile (Figure shown in Response 4.4).

Line 497 –

GC normalization

A GC-normalization step was performed to correct for differences in GC profile between samples. First, a GC profile was first determined for each downsampled BAM file. GC bias metrics were generated using Picard (version 2.19.0) CollectGcBiasMetrics with a WINDOW_SIZE of 300bp based on previous literature on GC bias in cfDNA³⁸. Next, GC profiles for all samples belonging to the same sequencer ID were loaded into R, and a generalized additive model (GAM) was used to generate an averaged profile for the batch, using ggplot geom_smooth() using method = 'gam' and formula 'y ~ s(x, bs = "cs")'. Extended Data Fig. 3 shows GC profiles for samples sequenced on two sequencers, plus the GAM averaged GC profile.

The averaged GC profile was used to normalize the mutation counts of all samples, based on the GC content of each mutated read as follows: a custom R script was used to annotate all mutations in each sample with their associated GC sequence content, rounded to the nearest 1%. The number of mutations in each GC content % bin was normalized relative to the averaged GC profile belonging to that sequencer, aiming to mitigate differences in GC-bias.

3. Fragment size analysis of mutant reads (Lines 133-139). Did the analysis that revealed a 2bp shorter length in cancer samples include all variants, both SNVs and SNPs?

a. The ~148bp fragment length for mutation reported sticks out as very discrepant from the expected 167bp fragment length of cfDNA. While this is likely because Pointy requires both R1 and R2 reads contain the mutation, it can be misleading if the reader is not aware of this detail. I would question and weigh whether this analysis is providing anything of value in light of its potential to be misleading.

4.6 Thank you for this feedback. This analysis showed a 2bp difference in mutant reads using SNP-subtracted data, i.e. SNVs only. Indeed, due to the requirement for both R1 and R2 reads to support a mutation in Pointy data, there is an inherent discrepancy in size compared to published literature (148bp vs. 167bp¹). This requirement for concordant reads effectively serves as a soft size selection step. Therefore, to improve the clarity of the manuscript, we have now removed the previous Lines 133-139.

4. In silico mutation fitting and performance evaluation. This is an important and necessary inclusion in the manuscript, so it is great to see this systematic evaluation. However, the methods section in the supplement is far too abstracted and details are simply glossed over, such that it is not easy to interpret or likely reproduce the results.

a. How were the 10-1000 mutations selected? What is the source?

4.7 Thank you for your positive feedback and suggestions on this *in silico* analysis. We have now clarified the source of the 10-1000 mutations that were spiked in, as follows:

Supplementary Methods –

Individual SBS signature spike-in for sensitivity and specificity of signature fitting

To assess the sensitivity of signature fitting to Pointy data, we performed an *in silico* signature spiking experiment. Varying numbers of mutations belonging to an individual SBS signature (between 10 and 1,000) were iteratively spiked into a randomly selected healthy mutation profile. One signature at a time was spiked in, to assess the recovery of each signature using signature fitting, and was repeated for each SBS in Alexandrov et al.¹. This was iteratively performed 100 times. In each iteration, prior to signature fitting, a background-subtraction step was performed, as described in the Methods.

A custom R script was used to generate spike-in signatures belonging to one signature only with a specified total number of mutations in the signature, utilizing countsSampling from the scRecover() package (v1.10.0). Fixed doses of specific mutational signatures were generated by sampling 10, 100 or 1,000 mutations (Extended Data Fig. 5b) with sampling frequencies equal to their frequency in reference mutational signatures³.

The contribution of each signature was assessed pre- and post- spike. This allowed the calculation of sensitivity and specificity of signature fitting based on the ratio of observed vs. expected increase in mutations. The expected number of mutations per signature equals either the 10, 100 or 1,000 mutations that were spiked in, plus those originally present in the control sample.

The factors influencing signature fitting sensitivity were assessed by comparing cosine similarity between signatures and ‘flatness’, as determined by the standard deviation of the proportion contributions of each SBS context within the signature.

During this analysis, we also updated the source of the control healthy SBS profile from a mean across all healthy samples to a randomly selected healthy sample (100 iterations). This allows the background-subtraction step to be performed on all other control samples, removing any circularity during this step. An updated Extended Data Fig. 5a is shown below.

Extended Data Fig. 5. *In silico* signature spike experiment to assess signature fitting sensitivity. (a) Using the PGDX plasma data, an experiment of *in silico* signature spike-in and assessment of signature fitting sensitivity was performed. Signature fitting sensitivity was defined as the ratio of the observed vs. expected increase in signature following spiking in known signatures to an averaged control SBS profile, with background-subtraction. Between 10-1000 mutations were spiked in. 100 iterations were used.

b. The definition of efficiency needs to be properly defined in the methods text—currently it is found in the captions of Extended Data Fig 5 and 6. What is the “expected increase” in signature in the efficiency definition and how is this defined? What is the ground truth signature contribution after spike-in and how is that defined?

4.8 Of course, we have now clarified the definitions of spike-in efficiency and ‘expected increase’ in signature contribution in the Supplementary Methods. We have also changed the terminology from efficiency to sensitivity for clarity. Please see the edited text below:

Supplementary Methods –

A custom R script was used to generate spike-in signatures belonging to one signature only with a specified total number of mutations in the signature, utilizing countsSampling from the scRecover() package (v1.10.0). Fixed doses of specific mutational signatures were generated by sampling 10, 100 or 1,000 mutations (Extended Data Fig. 5b) with sampling frequencies equal to their frequency in reference mutational signatures³.

The contribution of each signature was assessed pre- and post- spike. This allowed the calculation of sensitivity and specificity of signature fitting based on the ratio of observed vs. expected increase in mutations. The expected number of mutations per signature equals either the 10, 100 or 1,000 mutations that were spiked in, plus those originally present in the control sample.

c. Is the evaluation based on assessing the “contribution of signatures” or the “contribution of mutations to a given signature”? This is not clear. The contribution of signatures seems to be important but it’s not clear how the ground truth (expected contribution) is computed.

4.9 Indeed, we have amended the text to clarify that the aim of this experiment was to assess the recovery of an individual signature pre- and post-spike, as follows:

Supplementary Methods–

Individual SBS signature spike-in for sensitivity and specificity of signature fitting

To assess the sensitivity of signature fitting to Pointy data, we performed an *in silico* signature spiking experiment. Varying numbers of mutations belonging to an individual SBS signature (between 10 and 1,000) were iteratively spiked into a randomly selected healthy mutation profile. One signature at a time was spiked in, to assess the recovery of each signature using signature fitting, and was repeated for each SBS in Alexandrov et al.¹. This was iteratively performed 100 times. In each iteration, prior to signature fitting, a background-subtraction step was performed, as described in the Methods.

Regarding the definition of expected contribution, please see Response 4.8 above.

d. Was only a single signature spiked-in? What is the ability and performance of Pointy for estimating multiple signatures within a sample, which can often be the case in cancer.

4.10 Yes, Extended Data Figs. 5 and 6 showed the sensitivity and specificity of signature fitting following the spike-in of an individual signature into a control sample. To assess the performance of Pointy in the context of multiple signatures, we have now performed an *in silico* experiment whereby all signatures were spiked in alongside SBS1, to assess the effect of SBS1 co-spike on the observed increase in all other individually spiked-in signatures.

Additionally, to further characterize the ability of this method to resolve multiple signatures in parallel, we compared Pointy signatures in low-coverage WGS against signatures identified from plasma targeted sequencing (Extended Data Fig. 9). Please see Response 4.14 for this comparison and see below for the co-spike in experiment we have added.

Line 147 -

To assess the performance of signature recovery in the setting of multiple signatures, we iteratively spiked in signatures and simultaneously spiked in SBS1 at a ratio of 1:1 or 10:1 (Extended Data Fig. 5d, Supplementary Methods). At a 1:1 ratio of spike-in of both signatures, there was no impact on signature fitting. However, when 10x more SBS1

mutations were spiked in compared to the signature of interest, the rate of on-target signature fitting was reduced in multiple signatures (Benjamini-Hochberg corrected $p < 0.05$), especially signatures with low cosine similarity to SBS1 (linear regression $p = 1.5 \times 10^{-9}$). Signatures with similarity to SBS1 gained mutations directly from SBS1 ($q > 0.05$), whereas signatures with low similarity to SBS1 lost mutations, likely to other signatures gaining from SBS1. We show the extent of false positive signature fitting in the context of a singly spiked signature in Extended Data Fig. 6, where the proportion of mutations that were mis-attributed ranged from 1.7% with 10 mutations spiked, to 0.1% with 1,000 mutations spiked.

d

Extended Data Fig 5. In silico signature spike experiment to assess signature fitting sensitivity. (d) Co-spike experiment of SBS1 plus each signature at a ratio of 1:1 (left panel) or 10:1 (right panel). Baseline spike in sensitivity using 10 mutations is shown on the x-axis, and sensitivity with 1:1 or 10:1 SBS1 co-spike is shown on the y-axis. Signatures with zero mutations fitted with nil SBS1 spike-in are not shown. Data point size is proportional to the cosine similarity of that signature to SBS1. Co-spike in of each signature was repeated with 100 iterations with each setting. Signatures with a significant difference in sensitivity, following Benjamini-Hochberg correction, compared to the nil spike-in setting are highlighted in red.

e. Show example SBS profiles for spike-in's of 10, 100, 1000 to emphasize the difference in the input data signals for the number of mutations.

4.11 Absolutely, we have added the following panel as Extended Data Fig. 5b.

Extended Data Fig. 5. *In silico* signature spike experiment to assess signature fitting sensitivity. (b) Example SBS profiles for a background sample with 10-1,000 mutations spiked in, belonging to SBS1.

f. Extended Data Fig 5 caption lists Pearson $r=1 \times 10^{-12}$ instead of Pearson $r=0.71$, $p=1.6 \times 10^{-11}$.

g. Main text says “94% for 500 mutations but in Extended Data Fig 5a, it shows 10, 100, 1000 mutations.

4.12 Thank you for the above corrections (f) and (g). We have now corrected the Pearson correlation value in the caption for Extended Data Fig. 5. We have amended the main text to read as 1,000 mutations.

5. Signature calling thresholds. What *in silico* experiment does Line 187 refer to? How did the authors arrive at 13.9 median number of mutations (Extended Data Fig. 8) required to call a signature come? Please describe in the methods section.

4.13 The 95% detection thresholds were determined as the 95th percentile of each SBS signature contribution in healthy control individuals. Across all SBS signatures in the PGDX cohort, the median 95% detection threshold was 13.9 mutations. This was formerly plotted in Extended Data Fig. 8. However, on re-review of the manuscript, we have now removed this due to redundancy as it represents similar data to Extended Data Fig. 5, which shows the sensitivity for spiked-in signatures, which was both performed iteratively and performed for a variety of numbers of spiked-in mutations.

6. Classification of MSI vs MSS. The analysis using signature SBS20 to classify MSI CRC sample is not very convincing both for the low sample size, lack of validation cohort, technical issues with the machine learning procedure, and that there are likely other features more predictive of MSI.
a. According to the methods, 10-fold cross-validation was used. Since only 7 MSS samples are

available, even with stratified CV, some of the folds will not contain MSS samples. Likely this will result in over-fitting anyway because of the small sample size. The authors may want to consider leave-one-out CV (jack-knifing).

b. 16 CRC samples (11 MSI vs 5 MSS) is small and fine as a discovery cohort for this proof-of-concept. However, since this cohort has such small sample size, an independent validation cohort to test the trained model is required. The trained model on the 16 samples should be used to predict MSI status in a different test set cohort.

c. The usefulness of this model is trumped by simply using TMB. Aside from PGDX9805P1, all other MSI-H samples have high TMB.

4.14 We acknowledge the limitations of drawing conclusions from this small sample size for MSI classification, and thus we have edited the manuscript to avoid overstating these results. We have removed this classification experiment using ML given the risk of overfitting and low sample size. Instead, we compare MSI signature contribution levels between patients with/without MSI and controls (Extended Data Fig. 8). Therefore, this paragraph now reads:

Line 181 -

Aging signatures were detected in 10 out of 16 (62.5%) patients, and MSI signatures in 11 out of 16 (69.0%, Fig. 2e). Patients with MSI-H tumors had significantly greater SBS20 and SBS21 contributions than controls, whereas patients with MSS tumors were non-significantly different (Extended Data Fig. 8).

7. Cancer-relevant mutational signature prediction shows some interesting signals in the appropriate cancer types. It would be a stronger result if further confirmation came from deeper coverage sequencing where mutations can be called more accurately. For example, analyze signatures in a publicly available dataset of deeper whole genome sequencing and apply Pointy to the down-sampled version.

4.15 Indeed, to benchmark signature Pointy against gold-standard mutation calls, we obtained targeted sequencing calls for CRC plasma from the Georgiadis et al. paper², and signatures were fitted (Supplementary Methods). Both approaches identify aging and MSI signatures despite the different methodological approaches used (Extended Data Fig. 9). This analysis highlights that Pointy may be limited its ability to resolve between similar signatures, like due to limited sequencing depth. This would be improved with germline sequencing and deeper sequencing. We have added the following:

Extended Data – Supplementary Methods -

Mutational signature profiling and detection in targeted sequencing

Mutation calls (n = 170) generated by targeted sequencing were obtained for samples from the Georgiadis et al.² cohort (n = 15). Mutation calls were annotated with sequence context to generate a 96-SBS mutation profile for each sample. As mutation calls were already determined to be non-germline and above background noise, signatures were fitted directly, and a signature was detected if it had greater than zero mutations attributed to it.

Line 187 -

Signatures identified in CRC patient samples were compared against signatures fitted to targeted sequencing mutation calls on the same samples²¹ (Supplementary Methods). Both approaches identified aging and MSI signatures, with a 77.6% agreement across all signatures (Extended Data Fig. 9). Targeted sequencing identified SBS15 (Extended Data Fig. 9a), which was not detected with 95% specificity in Pointy data. We suggest that

SBS15 mutations may have been misattributed to SBS1 given their high cosine similarity (Extended Data Fig. 9b), and relatively low sensitivity for SBS15 from spike-in benchmarking (Extended Data Fig. 5a). When the cluster of similar signatures identified in Extended Data Fig. 9b (SBS1 and SBS6) were excluded from signature fitting, SBS15 could be observed (Extended Data Figs 9c-d). Germline subtraction and mutation calling would likely improve the resolution of signature profiling, although this would conventionally require 1-2 orders of magnitude greater sequencing^{16,20}.

Extended Data Figure 9. Comparison of signatures identified by Pointy and targeted sequencing. (a) Signatures identified using targeted sequencing of plasma samples in the CRC cohort (Methods). Detected signatures are indicated in red; non-detected signatures are indicated in blue. Samples that were not sequenced with targeted sequencing are indicated in gray. (b) Cosine similarities between signatures assessed in CRC samples, clustered by similarity. (c) Signatures identified using Pointy on low-coverage WGS (10M reads) on the same CRC plasma samples. (d) Signature detection with SBS1 excluded from signature fitting. (e) Signature detection with SBS1 and SBS6 excluded from signature fitting.

Signatures that became detected are indicated in yellow. (e) Signature detection with both SBS1 and SBS6 excluded from signature fitting.

We highlight the limitations of this low-coverage approach in the Discussion, and highlight the need for follow-up studies with greater sequencing depth in plasma, as follows:

Line 397 -

This study has notable limitations. We carried out this analysis with low-coverage WGS without matched germline samples, which improves the scalability of the approach but limits sensitivity for signature detection. This is particularly relevant for signature fitting, where the resolution for low-abundance and similar signatures was hampered by noise. Performing germline sequencing with low coverage sequencing alone would be of limited benefit: if 0.3x WGS (10M reads) were used to sequence a matched germline, only 13% of SNPs would be identified in the germline with 1 mutant read each. Germline calling conventionally requires >15x coverage^{16,20}, representing a 1-2 order of magnitude increase in sequencing, and would also need to be performed for healthy individuals to enable comparison. Future studies using deep sequencing in plasma and matched germline samples are needed to fully characterize circulating signatures.

8. Cancer detection analysis in CRC cohort.

a. Line 246 refers to xgboost and directs the reader to the Methods but it is not described in the Methods at all.

4.16 Thank you for highlighting this - we have now modified the methods to describe each of the machine learning models tested, as follows:

Line 558 -

xgboost (version 0.90.0.2) was used in R with the default parameters and nrounds = 100. randomForest (version v4.6-14) was used in R with the default parameters and ntree = 500. For SVM, svm() from the e1071 package (version 1.7.9) was used with default settings. For logistic regression, glm() from the stats package (version 4.1.2) was used with default settings. Following each iteration of cross-validation, a Pointy score for each sample was generated, ranging from 0 to 1 (higher represents more likely to be cancer). Classification performance characteristics were determined using the ci.cvAUC function from the cvAUC package (version 1.1.0) in R, using Pointy scores from all iterations as input. Random Forest showed the best performance (Extended Data Fig. 11) and was selected for use subsequently with nested 10-fold cross-validation with 500 iterations. Samples were classified using RF models using this approach for each sequencer within each study. Median Pointy scores are shown in Table 1. Pointy scores from all iterations from all samples from each study were used as input into ci.cvAUC() to generate AUC values by cancer type and stage (Fig. 3, Extended Data Fig. 15).

b. The small sample size of 16 CRC vs 20 healthy may suffer from overfitting, since standard 10-fold CV and only 50 iterations is performed. To properly evaluate the performance, the authors should use the trained model and predict on a different test set cohort. The Cristiano et al. DELFI CRC dataset would be ideal.

4.17 Thank you for this helpful feedback. We apologize for the imprecision in our description of the method: rather than using standard 10-fold cross-validation, nested 10-fold cross-validation was used (see Fig. 2 by Vabalas et al., 2019⁵), with 10 iterations (now increased to 500 when applied across the cohort, see Response 4.21). That is to say, the randomForest() function was

used to generate a new RF model on k-1 folds, which was tested on the kth fold, which was held-out from training, repeated k times. Indeed, while k-fold CV procedures can produce biased performance estimates with small sample sizes, nested CV approaches have been shown to be robust to limited sample size^{5,6}. We have amended the methods to clarify this first point:

Line 549 –

For sample classification to either healthy or cancer, we first tested multiple classification methods using a nested 10-fold cross-validation method⁵, repeated 10 times, using: xgboost, Random Forest (RF), Support Vector Machine (SVM) and Logistic Regression. Nested k-fold cross-validation uses a series of training, validation and test splits, and has been previously demonstrated to be robust to limited sample size^{5,6}.

Next, regarding testing and training on each of the cohorts - we identify evidence of batch effect affecting mutation profiles of controls between the two studies (Extended Data Fig. 16, below), which may be exacerbated by the small size of the PGDX cohort sample size. This difference in control mutation profiles exists despite using similar quality filters and GC-bias correction measures (Methods). Batch effects may have arisen from the different geographical sites at which samples were collected^{2,7}, particularly in the PGDX study, controls were purchased from BioIVT, whereas patients were recruited. As a result, generalization of its RF model to other studies' data may be challenging. Considering these batch differences, we pooled all healthy and CRC patient samples across the two studies to allow training across batches, using nested cross-validation. Please see results, below:

Line 379 -

Lastly, we assessed generalizability of this approach across cohorts, as patients with CRC were common to both cohorts (Supplementary Methods). We identified evidence of batch effect affecting SNP-subtracted mutation profiles of healthy controls between the two studies (Extended Data Fig. 16a), despite using quality filters and GC-bias correction. This may be due differences in sample collection location, as cases were collected across multiple academic sites, with controls in the former study sourced from a separate commercial site^{15,21}. To mitigate this, we pooled healthy and CRC patient samples across the two studies to allow training across batches. 10-fold nested CV was performed using RF, resulting in an AUC of 0.84 (Extended Data Fig. 16b).

Extended Data Fig. 16. Batch effects across studies and cancer detection across cohorts
(a) PCA of 96-SBS profiles of healthy individuals from each of the datasets used shows evidence of batch effect (PGDX, red; DELFI, blue). **(b)** To assess the generalizability of Pointy across cohorts, samples from healthy controls and patients with CRC were pooled between the two studies and classification was performed using an RF model with 10-fold nested CV, using 10 iterations.

Based on the above discussion, we have amended the terminology used for the studies used in Fig. 1 - we now call the studies directly by their names (PGDX and DELFI cohorts), rather than calling the DELFI cohort the validation set. We highlight the need for further validation in an updated Discussion.

Line 410 –

To mitigate noise, we leveraged machine learning for classification of samples and used quality filters (Methods). Differences in noise profile observed between control samples arising from independent studies highlight the importance of validation of this approach on a larger scale across multiple cohorts.

9. Analysis of mutant fragments in DELFI cohort.

a. Why is breast cancer not included in the Extended Data Fig 12d?

4.18 Thank you for identifying this, we have now included all DELFI patient samples in this analysis, and recalculated the short:long fragment ratios, shown below. With a larger sample size, now we identify shortening of cfDNA only, particularly in the cancer types with the highest ctDNA fractions (NSCLC and CRC). Please see below.

Line 309 -

The ratio between short (<150bp) to long mutant fragments (>150bp) was assessed as a quality control (Supplementary Methods). Patients with CRC or NSCLC showed significantly shorter fragments than healthy controls (corrected $p < 1 \times 10^{-7}$, Wilcoxon test, Extended Data Fig. 12d). There was a significant correlation between ichorCNA tumor

fraction and short:long fragment ratio (Pearson $r = 0.42$, $p = 5.1 \times 10^{-8}$). These findings of short ctDNA fragments are consistent with previous literature^{14,25}.

d

Extended Data Fig. 12. Signature and fragmentation profiling of the DELFI cohort. (d) The ratio of mutant fragments below vs. above 150bp was compared for each cancer type. Two-tailed Wilcoxon tests were used to compare short:long fragment ratios between samples, and p-values were BH corrected.

b. Is the observation in Extended Data Fig 12d partially explained by ctDNA fraction? CRC appear to have the highest tumor fractions and gastric & pancreatic likely have lower tumor fraction.

4.19 Indeed, following our analysis in Response 4.18, we identify a significant correlation between ichorCNA tumor fraction and short:long fragment ratio (Pearson $r = 0.42$, $p = 5.1 \times 10^{-8}$), which we have now included in the text (see Response 4.18).

This association is also reflected in the positive correlation between total plasma mutations (across all signatures) and ichorCNA fraction in NSCLC and CRC, which we have added to Extended Data Fig. 12, as follows:

Extended Data Fig. 12. Signature and fragmentation profiling of the DELFI cohort. (c) Pearson correlations between total number of mutations and ctDNA fraction (determined by ichorCNA) for all patients in the DELFI cohort. Significance testing was performed with BH-correction for multiple testing. The vertical dashed line indicates the detection threshold of 0.050 for ichorCNA based on 95% specificity in controls in this study.

10. Cancer detection analysis in DELFI cohort is not really considered an independent validation cohort for the purposes of the machine learning classification.

a. As mentioned in Comment #8b, a proper use of an independent validation cohort is to apply the trained model from the discovery cohort to this validation cohort.

4.20 Thank you for this suggestion. Please see Response 4.17 above, where we discuss this query.

b. For reporting cross-validation performance on the DELFI cohort is fine but model overfitting may still be a limitation, especially when only 50 iterations were performed. Thus, the AUC values are not likely sampled enough to represent the true distribution of values. If bootstrapping is to be used, then it should follow the procedure of drawing samples into the training set with replacement, which will lead to ~63% of samples for training. Furthermore, since the model training was carried out on this same dataset, performance should be based on evaluation of an independent dataset (which was not part of the cross-validation). That is, use the trained model to predict cancer in a different test set.

4.21 Absolutely – we have now re-run cancer detection in the DELFI cohort using 500 iterations of 10-fold nested cross-validation. We have updated the methods accordingly. This is across the expanded cohort of $n = 405$ samples ($n = 199$ patients, $n = 206$ controls). This results in an overall AUC of 0.94 (95% CI 0.94-0.94).

Fig. 6. Cancer detection across cancer types. Classification of samples from the DELFI cohort (199 patients and 206 controls) as either healthy or cancer was performed using a random forest model using 10-fold nested cross-validation with 500 iterations. This cohort consisted of stage I-IV NSCLC (n = 27), stage I-III breast cancer (n = 48), stage I-IV CRC (n = 27), stage I-IV, 0 and X gastric cancer (n = 27), stages I-IV ovarian cancer (n = 26), stage I-III pancreatic cancer (n = 34). Across all stages and cancer types, an overall AUC of 0.94 was achieved (95% CI 0.94-0.94). Classification performance by individual cancer type and stage is shown in Extended Data Fig. 16.

During the nested cross-validation, samples were split into training and test sets iteratively using the createFolds function to generate balanced samples in each fold. We have added the following to the Methods:

Line 554 -
 CreateFolds() from the caret package (version 6.0-90) was used to generate balanced folds for each round of cross-validation.

We acknowledge the importance of independent validation. Please see Response 4.17 for our discussion of our use of nested cross-validation and batch effects between datasets that limit training in PGDX and testing in DELFI.

c. How were the 137 cancer patient samples selected when there are 208 available? How were the 47 healthy donors selected when there are 215 available? Please describe in the Main text and Method section for the rationale to use a subset of samples with the same sequencing machine ID. How much of an impact does different sequencing machine for this dataset have on this approach? This seems to be quite important for your analysis, but no justification was provided. If GC-correction is already part of the approach to adjust batch effects, then this appears to be another potential artifact that has not been clearly shown for this datatype and its impact on the approach. Please provide data to support this.

4.22 Thank you for highlighting this. We have now expanded the cohort of DELFI samples analyzed to include 199 patients and 206 controls from the Cristiano et al.⁷ study. Two sequencing IDs were excluded due limited sample number, which we have indicated in Extended Data 12a

(see below). We are currently liaising with the EGA to download Bile Duct Cancer patient samples, which we could include in a subsequent revision, if our manuscript were of interest. We have amended the Methods to reflect the expanded cohort, as follows:

Line 431 –

Patient and sample characteristics

In this study, cfDNA WGS data were analyzed from a total of 215 patients and 227 healthy control individuals across two cohorts (Supplementary Table 1). For the initial cohort (PGDX), 16 patients with stage IV CRC and 21 healthy control individuals were recruited, consented and samples were collected as performed as described previously^{21,34}. We next studied 199 patients and 206 healthy control individuals from the DELFI¹⁵ dataset following approval from their Data Access Committee (DAC). Patients with the following cancer types were included from the DELFI study: CRC (n = 27), gastric (n = 27), NSCLC (n = 37), ovarian (n = 26), breast (n = 48), pancreatic (n = 34). Only pre-treatment timepoints from the DELFI study were used. For this proof-of-principle study, no blinding or randomization were performed.

Extended Data Fig. 12. Sample characteristics of the DELFI cohort and sequencer batch effects. (a) Samples were sequenced across multiple sequencers in the DELFI study. The number of samples per sequencer is shown by cancer type. For ML classification analyses, samples were controlled against healthy individuals sequenced on the same sequencer, based on sequencer ID. *Sequencers HWI-D00812, HWI-D00133R were excluded due to limited sample number and/or containing healthy samples only; duodenal cancer samples were excluded due to limited sample number. (b) PCA of 96-SBS mutation profile of healthy individuals sequenced across 4 sequencers in the DELFI cohort, following GC-correction, indicating differences in profile secondary to sequencer.

A GC-bias correction step was performed, analogous to Cristiano et al.⁷. This step mitigates differences in GC-bias between individual sequencing runs on the same sequencer (Extended Data Fig. 3). However, when applied across data generated from different sequencers, differences in SBS profile of control samples persist (Extended Data Fig. 12b). Therefore, classification analyses in this study were performed on samples run on the same sequencer. We have added the following text:

Line 273 -

Samples were sequenced across multiple sequencers (Extended Data Fig. 12a), leading to batch differences in SBS profile in healthy individuals despite correction for GC-bias (Extended Data Fig. 12b). Therefore, cases were compared against controls from the same sequencer.

Through expanding the cohort size to $n = 442$ (215 patients, 227 healthy) following your feedback, we now show:

- 1) Signature detection in plasma ranged from 84% in stage I-IV NSCLC to 40% in stage I-III pancreatic cancer (Fig. 4a) and ranged from a median of 61% detection in stage I-III disease to 78% in stage IV disease (Fig. 4b).
- 2) We are now better powered to detect physiological signatures in healthy individuals. We now detect a significant correlation between SBS1 in plasma and chronological in healthy individuals (Fig. 5b), suggesting the possibility of pre-cancer signature profiling in future.
- 3) We confirm an AUC of 0.94 for cancer detection using 10-fold nested cross-validation (repeated 500 times) in the expanded DELFI cohort across cancer types (Fig. 6). Please see Response 4.21 for further details.

Fig. 4. Signature profiling of the DELFI cohort. (a) The proportion of patients with signatures detected in plasma from the DELFI cohort are shown by cancer type. Signatures were detected at 95% specificity using 10M reads ($n = 199$, Methods). Error bars indicate standard error of the mean. (b) Proportion of patients with signatures detected in plasma, by cancer stage. Error bars indicate standard error of the mean.

Fig. 5. Profiling aging signatures in healthy individuals. (a) 159 healthy individuals' plasma WGS data (50M reads) from the Cristiano et al.¹⁵ study, sequenced on the same machine, were used to study the relationship between aging signatures in plasma and chronological age. Correlations between signatures were assessed with SNPs-retained, which showed a group of signatures that were significantly correlated with SBS1. Only correlations with a significance of $p > 0.05$ are shown in color. (b) SBS1 and SBS1-correlated signatures were tested for their association with aging. Following Benjamini-Hochberg correction, SBS1, SBS30 and SBS33 showed significant association with chronological age (all signatures tested are shown in Extended Data Fig. 14).

11. The link to the code does not work or files have been removed.

4.23 Thank you for highlighting this, we have now moved this to Bitbucket instead of Github, with the same login details (tested working - May 2022), as follows:

[Redacted]

I was also asked to evaluate the response to the concerns that Reviewer #2 raised. Please see the attached file for my comments on which concerns still need to be addressed.

The authors have added error-suppression to based on a similar approach by Zviran et al. However, MRDetect in Zviran et al. was working with much deeper genome sequencing data, which may not necessarily apply to low-coverage data. The authors should provide an evaluation of the benefits of this error-suppression approach, especially if no germline controls are included.

4.24 Please see Response 4.3 for a comparison between error-suppressed and non-error-suppressed data for cancer detection.

The description of using and subtracting SNPs in the analysis is more clear. However, overall, I still find that the description in the methods is not always clear, and details are not always sufficient. This is true not only for the pipeline but for other sections. Please see Comment 2b, 4b,c,d, 8a, 10c.

4.25 This query refers to comments 2b, 4b, 4c, 8a, 10c in the above feedback. Please see Responses 4.5, 4.8, 4.9, 4.16, 4.22.

We also note Reviewer #2 highlighted the use of fragment selection “in some analysis and not others”. In the previous revision, we had moved this analysis of size-selected signatures in healthy individuals to Extended Data, though we have now removed the use of fragment selection entirely to reduce ambiguity around the method.

I find that this (new) analysis for MSI classification is not very convincing. See my Comment #6.

4.26 Regarding MSI classification, please see Response 4.14 where we have scaled back this analysis and claims accordingly.

My concerns with this response:

1. CRC cancer detection from DELFI data alone showing cross-validation performance is not a true independent validation of the model developed on the original 16 CRC samples. The authors should use this DELFI CRC dataset to as a test dataset for the model trained in the original CRC+healthy dataset of 36 samples. See Comment 8b and 10a.

4.27 Thank you for this feedback, please see Response 4.17 for our discussion of cross-validation and batch effects across studies.

2. The authors should address the potential of overfitting for the cancer detection in this cohort. Since the performance reported is based on cross-validation and 50 iterations, overfitting may be an issue. See Comment 10b.

4.28 Please see Response 4.21, where we have increased the number of iterations for 10-fold nested cross validation to n = 500 iterations.

3. The selection of a subset of samples based on sequencing machine ID needs to be properly described and justified. See Comment 10c.

4.29 Please see Response 4.22 for description and justification of the analysis of samples by sequencer machine ID.

We would like to thank Reviewer #4 for taking the time to review our manuscript and providing helpful feedback, plus their review of Reviewer #2's feedback. As a result, we have strengthened this study by confirming our findings in a larger sample size, improved the precision of the Methods, and we have more fully characterized and discussed the limitations of this approach.

REVIEWERS' COMMENTS

Reviewer #4 (Remarks to the Author):

Technical comment:

Response 4.21 and Lines 362-369 --

95% confidence values have very narrow (often 0) ranges. The use of `ci.cvAUC()` appears to use influence-based confidence estimation. Traditionally, bootstrapping techniques as I had initially described in Comment 10b. The authors should double-check these 95% CI estimates because it appears odd, especially when some of the individual cancer types have such small sample sizes and the CI should be larger.

The authors have addressed all of my other concerns.

REVIEWERS' COMMENTS

Reviewer #4 (Remarks to the Author):

Technical comment:

Response 4.21 and Lines 362-369 --

95% confidence values have very narrow (often 0) ranges. The use of `ci.cvAUC()` appears to use influence-based confidence estimation. Traditionally, bootstrapping techniques as I had initially described in Comment 10b. The authors should double-check these 95% CI estimates because it appears odd, especially when some of the individual cancer types have such small sample sizes and the CI should be larger.

The authors have addressed all of my other concerns.

Thank you for your feedback. Indeed, we have now reviewed this and realize that using bootstrapped data as input into `ci.cvAUC()` leads to a falsely narrow estimate of confidence, as it is not designed for this purpose. We have now instead repeated this analysis by running `ci.cvAUC()` 500 times, once on each iteration, then take the 2.5th and 97.5th percentiles of the resulting bootstrapped AUCs to generate a bootstrapped 95% CI. The spread of the AUCs in Fig. 6, and Supplementary Figs. 11 and 15 are now more realistic. To highlight the spread of the data, we now show all bootstrapped ROCs. Please see updated ROC figures and their updated 95% confidence intervals below.

Fig. 6. Cancer detection across cancer types. Receiver Operative Curve (ROC) of classification of samples from the DELFI cohort, which included 199 individuals with cancer, and 206 healthy individuals, was performed. A random forest model using 10-fold nested cross-validation with 500 iterations was used to classify samples as either healthy or cancer. This cohort consisted of stage I-IV NSCLC (n = 27); stage I-III breast cancer (n = 48); stage I-IV CRC (n = 27); stages I-IV, stage 0 and stage X gastric cancer (n = 27); stages I-IV ovarian cancer (n = 26); stage I-III pancreatic cancer (n = 34). All 500 ROC curves are shown. Classification performance

Response to reviewers

by individual cancer type and stage is shown in Supplementary Fig. 15. AUC, area under the curve.

Supplementary Fig. 11. Performance comparison of data processing steps and machine learning models. 0.3x plasma WGS data from healthy individuals ($n = 19$) and patients with stage IV CRC ($n = 16$) were used to test different machine learning models for classification using point mutations and copy number from ichorCNA as input (Methods). **(a)** Classification performance using xgboost with raw SBS mutation matrix input following SNP-subtraction. Nested 10-fold cross-validation was performed, repeated 10 times. AUC, area under the curve. **(b)** Classification performance using xgboost with PCA-transformed input following SNP-subtraction. **(c)** Classification performance using a logistic regression model with PCA-transformed input following SNP-subtraction. **(d)** Classification performance using a random forest model with PCA-transformed input following SNP-subtraction. **(e)** Classification performance using a support-

Response to reviewers

vector machine model with PCA-transformed input following SNP-subtraction. **(f)** Classification performance using a random forest model with PCA-transformed input following SNP-subtraction, with ichorCNA ctDNA fractions included in model training. **(g)** To test the effect of downsampling data to 10M reads, using a random forest model, sequencing data were iteratively downsampled (50 iterations) to 10M reads and classified into cancer (n = 16) vs. healthy (n = 19). **(h)** Classification performance using a random forest model with PCA-transformed, without SNP-subtraction. **(i)** Classification performance using a random forest model with PCA-transformed data, with SNP-subtraction, but without error-suppression (see Supplementary Fig. 11d for error-suppressed data). AUC, area under the curve; CV, cross-validation; PCA, principal component analysis; RF, random forest.

Response to reviewers

Response to reviewers

Supplementary Fig. 15. Classification performance using Pointy on the DELFI data set. Cancer detection performance using a random forest (RF) model in the DELFI data set was assessed across all cancer stages (n = 199), using 10-fold nested cross-validation using a random forest model (500 iterations). (a) stage I-IV non-small cell lung cancer (NSCLC, n = 37). (b) stage I-III breast cancer (n = 48). (c) stage I-IV CRC (n = 27). (d) stage I-IV and X gastric cancer (n = 27). (e) stages I, III and IV ovarian cancer (n = 26). (f) stage I-III pancreatic cancer (n = 34). (g-j) Detection rates were next assessed by stage: stage I (n = 41), stage II (n = 86), stage III (n = 33), stage IV (n = 36). (k) Detection rates by stage and cancer type for all patients and all stages (n = 199), using a 95% specificity threshold for detection. Healthy samples (n = 206) are included as stage = NA. Boxplots represent bootstrapped median, bottom and upper quartiles, and whiskers correspond to 1.5× IQR. Points indicate outliers. (l) PCA of plasma SBS mutation profiles from patients all sequenced on the same sequencer (HWI-D00837, n = 70). This shows separation of samples by cancer type in PC1 and PC2, which may enable classification by cancer type. AUC, area under the curve; CV, cross-validation; PC, principal component.

We again thank Reviewer #4 for their critical appraisal and feedback on our manuscript.